# The hippocampus supports deliberation during value-based decisions

**Akram Bakkour[1]\*, Daniela J Palombo[2†], Ariel Zylberberg[3,4,5], Yul HR Kang[3‡], Allison Reid[2], Mieke Verfaellie[2], Michael N Shadlen[3,4,5,6], Daphna Shohamy[1,4,6]**

[1]Department of Psychology, Columbia University, New York, United States; [2]Memory Disorders Research Center, VA Boston Healthcare System and Boston University School of Medicine, Boston, United States; [3]Department of Neuroscience, Columbia University, New York, United States; [4]Mortimer B Zuckerman Mind Brain Behavior Institute, Columbia University, New York, United States; [5]Howard Hughes Medical Institute, Chevy Chase, United States; [6]The Kavli Institute for Brain Science, Columbia University, New York, United States

**Abstract** Choosing between two items involves deliberation and comparison of the features of each item and its value. Such decisions take more time when choosing between options of similar value, possibly because these decisions require more evidence, but the mechanisms involved are not clear. We propose that the hippocampus supports deliberation about value, given its well-known role in prospection and relational cognition. We assessed the role of the hippocampus in deliberation in two experiments. First, using fMRI in healthy participants, we found that BOLD activity in the hippocampus increased as a function of deliberation time. Second, we found that patients with hippocampal damage exhibited more stochastic choices and longer reaction times than controls, possibly due to their failure to construct value-based or internal evidence during deliberation. Both sets of results were stronger in value-based decisions compared to perceptual decisions.
DOI: https://doi.org/10.7554/eLife.46080.001

**\*For correspondence:**
ab4096@columbia.edu

**Present address:** [†]Department of Psychology, The University of British Columbia, Vancouver, Canada; [‡]Computational and Biological Learning Lab, Department of Engineering, University of Cambridge, Cambridge, United Kingdom

**Competing interests:** The authors declare that no competing interests exist.

## Introduction

Some decisions involve more deliberation than others. Even seemingly simple decisions such as those that involve preferences between a pair of familiar items take more time when they involve a choice between options of similar subjective value. This simple observation holds across many kinds of decisions, whether they are based on perception of the environment—is the apple green or red? (*Cassey et al., 2013*; *Gold and Shadlen, 2007*; *Ratcliff, 2002*; *Usher and McClelland, 2001*)—or on internal values and preferences—do I prefer a green apple or a red one? (*Basten et al., 2010*; *Hunt et al., 2012*; *Krajbich et al., 2010*; *Milosavljevic et al., 2010*). One explanation for why such decisions take more time is that a commitment to a choice depends on the accumulation of evidence to a threshold, and when the evidence is weaker, more samples are required to reach such a threshold (*Krajbich et al., 2010*; *Milosavljevic et al., 2010*). This idea has been studied extensively in perceptual decisions about dynamic stimuli (e.g. moving dots) for which more time clearly provides more samples of external evidence, and therefore can improve the accuracy of the decision (*Britten et al., 1996*; *Britten et al., 1993*; *Hanks et al., 2015*; *Mazurek et al., 2003*; *Newsome and Paré, 1988*; *Salzman et al., 1990*). It is less clear why the same framework would apply to value-based decisions, which depend on internal evidence (*Krajbich et al., 2010*; *Milosavljevic et al., 2010*). In such cases, it is not known what the source of the evidence is and why more samples should be required to decide between options that are close in value.

We sought to understand the processes involved in deliberation when making value-based decisions. Our central hypothesis is that the hippocampus plays a key role in this deliberation process, contributing to the comparison between items and the construction of internal samples of evidence bearing on the decision.

This hypothesis is guided by several observations. First, extensive research demonstrates that the hippocampus is necessary for detailed and vivid prospection about future events (*Addis and Schacter, 2008*; *Buckner, 2010*; *Hassabis et al., 2007*; *Klein et al., 2002*; *Race et al., 2011*; *Schacter et al., 2007*). This sort of prospection is likely to guide value-based decisions because it allows a decision-maker to imagine the detailed outcome of each choice option. Second, and more broadly, the hippocampus is known to contribute to relational encoding (*Cohen and Eichenbaum, 1993*; *Horner and Burgess, 2013*), a term coined by *Cohen and Eichenbaum (1993)* to capture the essential role of the hippocampus across many cognitive processes that involve flexible comparison and association between distinct items and features (for reviews, see *Barry and Maguire, 2019*; *Davachi, 2006*; *Eichenbaum, 2000*; *Eichenbaum, 2018*; *Konkel and Cohen, 2009*; *Palombo et al., 2015a*; *Shohamy and Turk-Browne, 2013*). This relational function of the hippocampus is thought to underlie its well-known role in episodic memory, but the comparison of multiple dimensions of items and their relation to each other is also likely to help guide deliberation during decision making by supplying internal evidence about each option. Recent studies have indeed linked hippocampal-based mnemonic processes to choice behavior by demonstrating that the hippocampus is involved in decisions that explicitly depend on memory by requiring participants to use novel associations acquired in the experiment (*Barron et al., 2013*; *Gluth et al., 2015*; *Wimmer and Shohamy, 2012*). However, a critical open question remains about whether the hippocampus also contributes to seemingly simple decisions—between two highly familiar items—without the explicit demand to use memory.

We conducted two experiments to address this question. First, we conducted an fMRI study in healthy young participants while they made decisions based on well-established subjective value (fMRI; Experiment 1). We reasoned that if the hippocampus supports deliberation, then longer decision times should be related to more engagement of the hippocampus. Second, to test whether the hippocampus plays a causal role in resolving value-based decisions, we tested amnesic patients with damage to the hippocampus and surrounding medial temporal lobe (MTL) as well as age-, education-, and verbal IQ-matched healthy controls (Patients; Experiment 2). Although a choice between two familiar items is not typically thought to depend on the hippocampal memory system (*Bartra et al., 2013*; *Padoa-Schioppa and Assad, 2006*; *Platt and Plassmann, 2014*; *Rangel and Clithero, 2014*; *Rangel et al., 2008*), we reasoned that amnesic patients may nonetheless show differences in the way they deliberate about simple value-based decisions. Amnesic patients could take *less time* because their decisions involve less deliberation, or they could take *more time* because they try unsuccessfully to deliberate using evidence derived from relational mechanisms. In the latter case, the extra time would not improve their decisions.

In both experiments, participants performed a *value-based decision* task in which they made a series of choices between two familiar food items (*Figure 1*). The subjective value of each individual item was determined for each participant using an auction procedure in advance (see Materials and methods), so that we could systematically vary the difference in value between the two items (i.e. ΔValue) during the decision task (see also *Grueschow et al., 2015*; *Krajbich et al., 2010*; *Milosavljevic et al., 2010*; *Polanía et al., 2015*). The same participants also took part in a control condition in which they made *perceptual decisions* about the dominant color of a dynamic random dot display (*Figure 1* and *Figure 1—video 1*). The perceptual comparison task solicits the same choice and reaction time behavior but is based on external sensory input.

In Experiment 1, we found that decision time in the value-based decision task was longer when the choice options were closer in value, as expected (*Krajbich et al., 2010*; *Milosavljevic et al., 2010*; *Polanía et al., 2015*). We also found that reaction times correlated with hippocampal BOLD activity, and this effect was localized to regions of the hippocampus that showed activity related to memory retrieval, independently identified in the same participants. In Experiment 2, we found that amnesic patients were somewhat more stochastic and much slower when making value-based decisions. Importantly, despite parallel behavioral findings in value-based decisions and perceptual decisions in the healthy controls, both the hippocampal BOLD effects and the impairments in patients

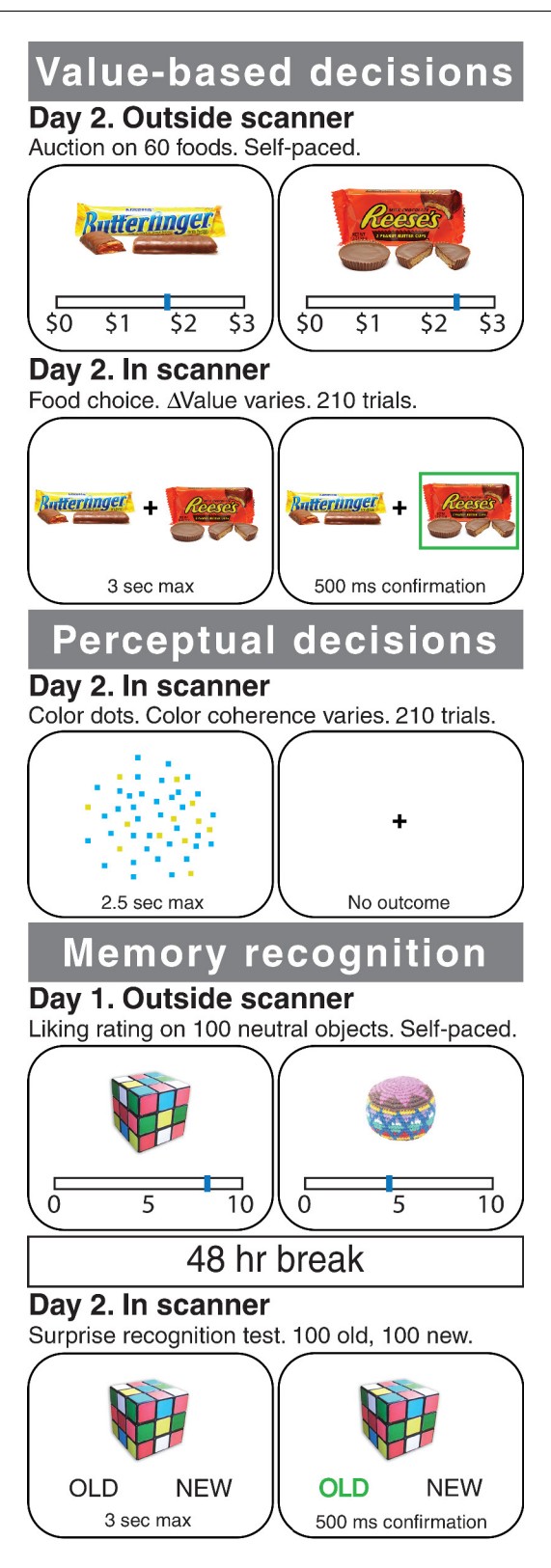

**Figure 1.** Experimental tasks. In Experiment 1, healthy participants were scanned with fMRI during three different tasks: a value-based decision task (top), a perceptual decision task (middle), and a memory recognition task (bottom). In the value-based decision task, participants were presented with 150 pairs of foods that differed on ΔValue (based on a pre-task auction procedure for rating the items; see Materials and methods). Participants were told to choose the item that they preferred and that their choice on a randomly selected trial would be honored at the end of the experiment. In the

*Figure 1 continued on next page*

*Figure 1 continued*

perceptual decision task, participants were presented with 210 trials of a cloud of flickering blue and yellow dots that varied in the proportion of blue versus yellow (color coherence). Participants were told to determine whether the display was more blue or more yellow. In the recognition memory localizer task, participants underwent a standard recognition task using incidental encoding of everyday objects: first, they rated 100 objects (outside of the scanner); 48 hr later they were presented with a surprise memory test in the scanner, in which 'old' objects were intermixed with 100 'new' objects, one at a time, and participants were asked to indicate whether each object was 'old' or 'new'. In Experiment 2, amnesic patients with MTL damage and healthy controls performed variants of the value-based and perceptual decision tasks (see Materials and methods).

DOI: https://doi.org/10.7554/eLife.46080.002

The following video is available for figure 1:

**Figure 1—video 1.** Video of the colored dots stimulus.

DOI: https://doi.org/10.7554/eLife.46080.003

were selective to the value-based decision task. Together, these findings establish a critical role for the hippocampus in value-based decisions about familiar choice options.

## Results

We conducted two experiments to test the mechanisms underlying deliberation in value-based decisions. In the first experiment, we scanned healthy young participants with functional MRI while they performed value-based and perceptual decision tasks. In the second experiment, we tested behavior in amnesic patients with damage to the hippocampus and surrounding MTL as well as age-, education-, and verbal IQ-matched healthy control participants on slightly modified versions of these two decision tasks (see Materials and methods).

### Experiment 1: functional MRI

#### Behavior in both decision tasks conforms to sequential sampling models

On the perceptual decision task, healthy young participants (n = 30) made more accurate decisions when the color was more biased toward blue or yellow (*Figure 2A*, top) and reaction times (RT) were longer for decisions between options that were more difficult to discriminate (i.e. color coherence near zero, *Figure 2A*, bottom). Similarly, on the value-based decision task, participants made decisions more consistent with their subjective valuation when $\Delta Value$ was larger (*Figure 2B*, top). RTs were longer for decisions between options for which the magnitude of $\Delta Value$ ($|\Delta Value|$) was smaller (*Figure 2B*, bottom). For both the perceptual and the value-based tasks, choices and RT were well described by drift diffusion models (*Figure 2*, solid lines). This observation is consistent with prior work (*Krajbich et al., 2015*; *Ratcliff and McKoon, 2008*; *Shadlen and Kiani, 2013*) and with the proposal that both types of decisions arise through a process of sequential sampling that stops when the accumulation of evidence satisfies a threshold or bound. The choice functions and range of RT were comparable in the two tasks, as were the goodness of fits (for model parameter estimates, see *Figure 2—source data 1*; for individual participant fits, see *Figure 2—figure supplement 1*). Some of the differences between the fits, apparent by eye, are attributed to the different scales of evidence strength in the two tasks (see *Figure 2—figure supplement 2*). We considered simpler parameterizations of the model, but the full model presented here produced a better fit compared to a model with no power law (BIC = 19.45), and a better fit compared to a model with no power law and flat bounds (BIC = 168.45).

#### Timing of value-based decisions is related to brain correlates of memory

We first conducted a whole-brain analysis to identify regions in the brain that show (*i*) an effect of RT: a correlation between RT and BOLD activity for the value-based task more so than for the perceptual task, and (*ii*) a memory effect: greater BOLD activity for successful retrieval of object memories (using the separate object-memory localizer task, see Materials and methods, *Figure 3—figure supplement 1* and *Figure 3—source data 1*). Each of these analyses of the fMRI data (RT; memory retrieval) identified largely separate networks of brain regions (*Figure 3—figure supplement 1* and Figure 3—figure supplement 3; *Stark and Squire, 2001*; *Yarkoni et al., 2009*). Critically, however, both showed significant effects in the hippocampus and, as shown in *Figure 3* (and *Figure 3—source data 2*), the *conjunction* of these two effects revealed significant shared BOLD activity in the

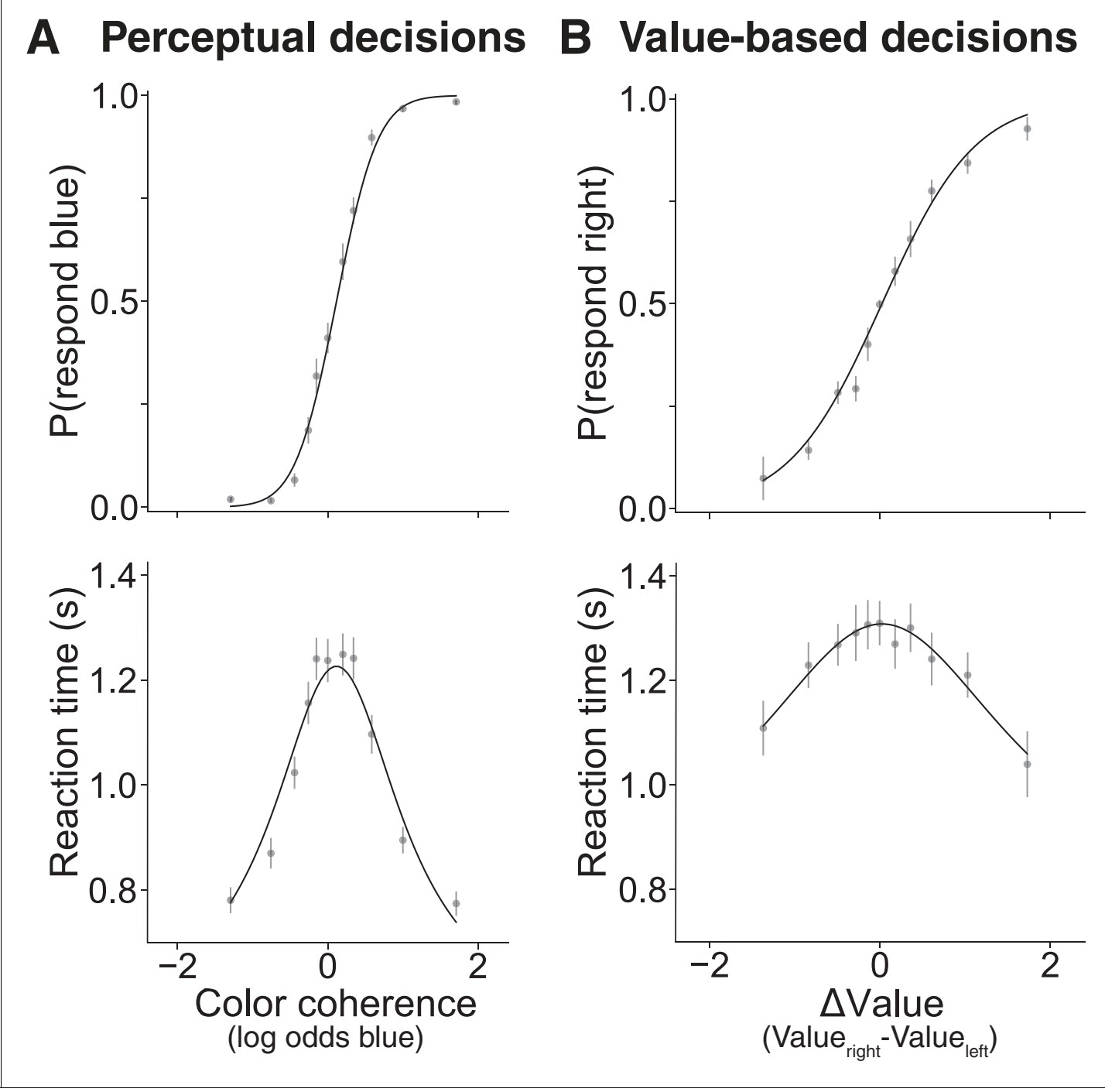

**Figure 2.** Choices between options that are similar take more time for both perceptual and value-based decisions in Experiment 1. Behavioral results from 30 young healthy participants for (**A**) perceptual and (**B**) value-based decisions. (**A**) Proportion of blue choices (top) and mean RT (bottom) plotted as a function of signed color coherence (the logarithm of the odds that a dot is plotted as blue). (**B**) Proportion of right item preference (top) and mean RT (bottom) plotted as a function of value difference (the subjective value of the item on the right side of the screen minus the subjective value of the item on the left) binned into eleven levels. Gray symbols are means (error bars are s.e.m.); solid black lines are fits to drift diffusion models. See *Figure 2—figure supplement 1* for fits to data from individual participants. See *Figure 2—figure supplement 3* for parameter recovery analysis.
DOI: https://doi.org/10.7554/eLife.46080.004

The following source data, source code and figure supplements are available for figure 2:

**Source code 1.** Jupyter notebook with analysis code and output for analyses performed on data from Experiment 1.
DOI: https://doi.org/10.7554/eLife.46080.008

*Figure 2 continued on next page*

*Figure 2 continued*

**Source data 1.** Parameter estimates and goodness of fit measures for Experiment 1.

DOI: https://doi.org/10.7554/eLife.46080.009

**Source data 2.** Trial-level data for the perceptual task in Experiment 1.

DOI: https://doi.org/10.7554/eLife.46080.010

**Source data 3.** Trial-level data for the value-based task in Experiment 1.

DOI: https://doi.org/10.7554/eLife.46080.011

**Figure supplement 1.** Data and fits for value-based and perceptual decisions per participant in Experiment 1.

DOI: https://doi.org/10.7554/eLife.46080.005

**Figure supplement 2.** Comparison of data and fits from *Figure 2* after rescaling the units of evidence.

DOI: https://doi.org/10.7554/eLife.46080.006

**Figure supplement 3.** Parameter recovery analysis.

DOI: https://doi.org/10.7554/eLife.46080.007

hippocampus. BOLD activity in memory-related hippocampal regions was more positively correlated with RT for value-based decisions than perceptual decisions, consistent with our hypothesis that deliberation associated with resolving preference relies on memory-related hippocampal mechanisms.

We conducted a series of control analyses to consider possible alternative explanations for the differential hippocampal activation on value-based versus perceptual tasks. First, the hippocampal BOLD activity might be related simply to the fact that the value-based decision task makes more demands on memory because it depends on identifying objects. Indeed, a main effect of value-based versus perceptual decisions reveals differences in BOLD activity along the ventral stream and in the medial temporal lobe, including the hippocampus (*Figure 3—figure supplement 2A* and *Figure 3—source data 3*). However, if object identification were the reason for the RT effects, one would expect to find only a main effect of task—that is, an overall difference between the two tasks regardless of deliberation time—rather than a significant interaction between task and RT. The observation of both a main effect of task and an interaction with RT suggests that differences in object recognition do not account for the finding in the hippocampus. Second, we wondered whether the hippocampal BOLD activity in the value-based task could be related to the fact that for some participants there was a difference in the range of RT in the value-based task compared to the perceptual task. To test this, we repeated the analysis using only trials that shared the same range of RT on the two tasks (by participant). This analysis revealed a similar result (*Figure 3—figure supplement 2B* and *Figure 3—source data 4*), suggesting that the difference in the hippocampus is not related to differences in RT range.

A third possibility we considered was that the tasks differ in overall levels of difficulty. Indeed, RT is a function of the difficulty levels in each of the two tasks, but there is also variability in RT within each level of difficulty, allowing us to address questions about RT while controlling for difficulty. Therefore, we tested the possibility that difficulty accounted for more of the variance in hippocampal BOLD activity than RT by repeating the same analysis as in *Figure 3* while controlling for the magnitude of color coherence and $\Delta$*Value*, as well as other potential correlates of RT (e.g. mean of the pair of values; see Materials and methods). This analysis again revealed RT-related activity in the hippocampus that is greater for value-based than perceptual decisions, even after accounting for other correlates of RT, both within an anatomical ROI of bilateral hippocampus and at a whole-brain corrected level (*Figure 3—figure supplement 3* and *Figure 3—source datas 5–8*). The conjunction between the RT effect and the memory map was again found within the hippocampus ROI (*Figure 3—figure supplement 3H*). Finally, because our memory encoding task involved value judgments (see Materials and methods), we reran the conjunction analysis using an independent memory recognition localizer that was not specific to value-based encoding, instead using two independent meta-analysis maps from neurosynth.org based on the terms 'autobiographical memory' and 'recollection'. The three-way conjunction between the differential effect of RT on BOLD and these two meta-analysis maps also shows overlap in the hippocampus (*Figure 3—figure supplement 4*).

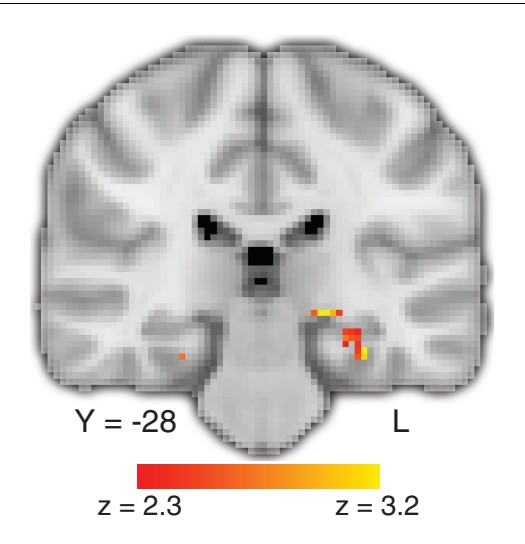

**Figure 3.** Deliberation time during value-based decisions is related to activation in the hippocampus. The figure shows a representative slice at the level of the hippocampus. The map exploits all three tasks and shows a comparison of the effect of trial-by-trial RT on value-based decisions with perceptual decisions, localized (with a conjunction analysis) to regions of the brain that also show a memory-retrieval effect. The full map can be viewed at https://neurovault.org/collections/BOWMEEOR/images/56727. This effect in the hippocampus was replicated with a separate analysis controlling for potential confounds (e.g. mean value across items in a pair; *Figure 3—figure supplement 3D*). Coordinates reported in standard MNI space. Heatmap color bars range from z-stat = 2.3 to 3.2. The map was cluster corrected for familywise error rate at a whole-brain level with an uncorrected cluster-forming threshold of z = 2.3 and corrected extent threshold of p<0.05.

DOI: https://doi.org/10.7554/eLife.46080.012

The following source data and figure supplements are available for figure 3:

**Source data 1.** Activation table for map in *Figure 3—figure supplement 1*; successful memory retrieval: hits > correct rejections.

DOI: https://doi.org/10.7554/eLife.46080.019

**Source data 2.** Activation table for map in *Figure 3*; conjunction between RT effect on BOLD for value-based greater than perceptual with effect of successful memory recognition.

DOI: https://doi.org/10.7554/eLife.46080.018

**Source data 3.** Activation table for map in *Figure 3—figure supplement 2A*; overall main effect of value-based greater than perceptual decisions.

DOI: https://doi.org/10.7554/eLife.46080.020

**Source data 4.** Activation table for map in *Figure 3—figure supplement 2B*; the effect of RT on BOLD for value-based greater than perceptual decisions,

*Figure 3 continued on next page*

## Connectivity between hippocampus and parietal cortex increases with value-based decision time

The fMRI results suggest that BOLD activity in the hippocampus is related to the time it takes to make value-based decisions. We next explored the broader neural circuits that interact with the hippocampus during value-based decisions and how activity in such circuits varies with RT. We used a psychophysiological interaction (PPI) analysis to identify brain regions with activity that covaried in an RT-dependent manner with the activity of hippocampal 'seed' voxels—that is those that exhibited RT-dependent activation on the value-based decision task and memory-related activation on the memory localizer task. The strongest RT-dependent correlation was between the hippocampus and the parietal cortex (superior parietal lobule and precuneus), showing that functional connectivity between the hippocampus and parietal cortex was greater for value-based decisions that took longer (*Figure 4* and *Figure 4—source data 1*).

## Experiment 2: behavior in amnesic patients

The fMRI data reveal that the timing of value-based decisions is related to BOLD activity in the hippocampus, suggesting a possible role for the hippocampus in the deliberation process. However, fMRI can only tell us about brain activity correlated with a mental process, leaving open the critical question of whether the hippocampus plays a direct, causal role in value-based decisions. Experiment 2 was designed to address this question by testing value-based decision making in patients with amnesia subsequent to damage to the hippocampus and nearby MTL structures.

Our overarching hypothesis is that the hippocampus contributes to value-based decisions by supporting the comparison of options, the simulation of outcomes, and the recollection of internal evidence. We therefore expected that damage to the hippocampus would impair this deliberation process. As noted earlier, we had no strong prediction regarding whether patients would show faster or slower RTs in general. We reasoned that slower RTs might reflect efforts to search for evidence to resolve decisions, whereas faster RTs might reflect choices that lack deliberative reasoning altogether. Patients with hippocampal damage are not known to have general impairments in valuation processes and the experiment only included food items that each patient fully recognized (see

*Figure 3 continued*

restricted to trials for which the range in RT was matched between the two decision tasks.

DOI: https://doi.org/10.7554/eLife.46080.021

**Source data 5.** Activation table for map in *Figure 3—figure supplement 3A*; effect of value-based RT on BOLD.

DOI: https://doi.org/10.7554/eLife.46080.022

**Source data 6.** Activation table for map in *Figure 3—figure supplement 3B*; effect of perceptual RT on BOLD.

DOI: https://doi.org/10.7554/eLife.46080.023

**Source data 7.** Activation table for map in *Figure 3—figure supplement 3C*; value-based RT > perceptual RT.

DOI: https://doi.org/10.7554/eLife.46080.024

**Source data 8.** Activation table for maps in *Figure 3—figure supplement 3E*; *Figure 3—figure supplement 3F*; *Figure 3—figure supplement 3G*.

DOI: https://doi.org/10.7554/eLife.46080.025

**Source data 9.** Activation table for map in *Figure 3—figure supplement 5*: Modulated effect of the value of the chosen food.

DOI: https://doi.org/10.7554/eLife.46080.026

**Figure supplement 1.** Parametric map of main effect of hits versus correct rejections during memory recognition.

DOI: https://doi.org/10.7554/eLife.46080.013

**Figure supplement 2.** Control analyses to consider alternative explanations for the differential hippocampal activation on value-based versus perceptual tasks.

DOI: https://doi.org/10.7554/eLife.46080.014

**Figure supplement 3.** Deliberation time during value-based decisions is related to activation in the hippocampus using a more complex model.

DOI: https://doi.org/10.7554/eLife.46080.015

**Figure supplement 4.** Timing of value-based decisions is related to activation in memory-localized regions of the hippocampus.

DOI: https://doi.org/10.7554/eLife.46080.016

**Figure supplement 5.** Value-coding brain regions.

DOI: https://doi.org/10.7554/eLife.46080.017

Materials and methods). Therefore, we expected that patients would make choices largely consistent with their subjective valuations. Finally, for the perceptual task, we expected the patients to show intact performance, consistent with the notion that the hippocampus is not needed to make decisions based on external evidence.

## Timing of value-based decisions is impaired in amnesic patients

We tested six amnesic patients with damage to the hippocampus and surrounding MTL on the decision tasks from Experiment 1, slightly modified to accommodate the patient population (see Materials ans methods). The patients have well-characterized memory impairments combined with intact verbal reasoning and IQ (see *Table 1*), and have participated in several prior studies (*Foerde et al., 2013*; *Grilli and Verfaellie, 2016*; *Palombo et al., 2019*; *Palombo et al., 2015b*). We compared the patients to fourteen age-, education-, and verbal IQ-matched healthy participants.

On the perceptual decision task, both patients and healthy participants made more accurate decisions when the color was more strongly biased toward blue or yellow (*Figure 5A*, top). The RTs of both the patients and healthy participants were longer for decisions between options that were more difficult to discriminate (i.e. color coherence near zero, *Figure 5A*, bottom). Patients took about the same amount of time as healthy controls to make a perceptual decision and there were no significant differences between the groups on accuracy (i.e. slopes of the choice function in *Figure 5A*, p=0.28) or RT (interaction between | color coherence| and group on RT, p=0.18; and main effect of group on RT, p=0.41). Further, for both groups, choices and RTs were well-described by a drift diffusion model (*Figure 5A*, solid lines), suggesting that damage to the hippocampus did not impair the patients' ability to make decisions that require sequential sampling of external evidence.

In contrast, on the value-based decision task the amnesic patients' performance diverged from that of healthy controls. Although the amnesic patients' choices were clearly governed by ΔValue (red sigmoid function, *Figure 5B* top, simple effect of ΔValue on choices among amnesics, p<0.0001), their choices were more stochastic than those of the controls (flatter red sigmoid function, *Figure 5B* top, p=0.0008). This observation implies that the amnesic patients were not randomly guessing or forgetting the subjective value of the items but were less sensitive to their difference. Notably, the patients did not show any obvious differences in their use of the value rating scale nor in the resulting range of ΔValues (*Figure 5—figure supplement 1*). This implies that the flatter choice function is not explained by a difference in the use of the value rating scale but that the ΔValue derived from that scale had less purchase on their choices.

The more striking difference between the two groups was observed on RT during value-based decisions: the amnesic patients were substantially slower than healthy controls (*Figure 5B* bottom,

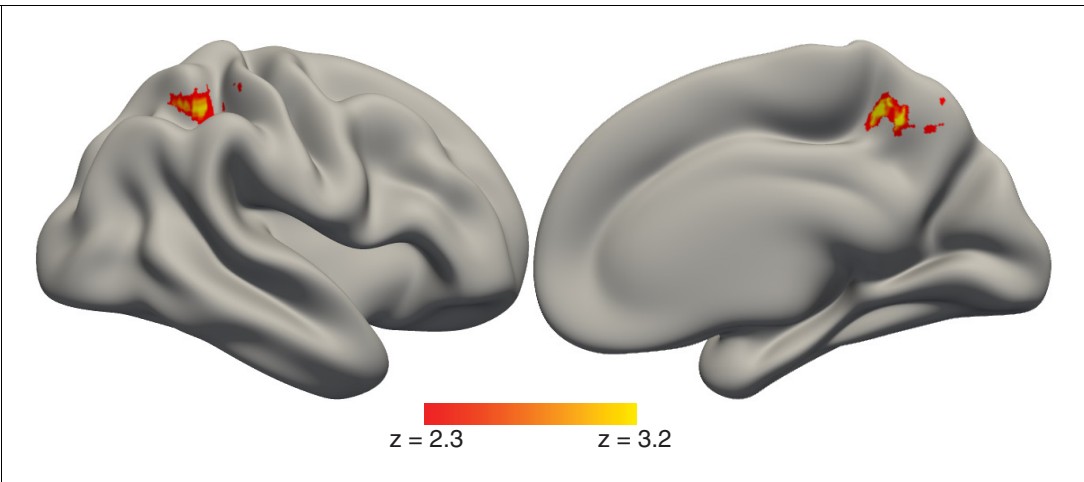

**Figure 4.** Timing of value-based decisions is related to functional coupling between the hippocampus and parietal cortex. Lateral (left) and medial (right) view of a semi-inflated surface of a template brain. PPI results were projected onto the cortical surface. There was a stronger correlation in activity between the hippocampus and the parietal cortex when value-based decisions took more time. The full map can be viewed at https://neurovault.org/collections/BOWMEEOR/images/129376. Heatmap color bars range from z-stat = 2.3 to 3.2. The map was cluster corrected for familywise error rate at a whole-brain level with an uncorrected cluster-forming threshold of z = 2.3 and corrected extent of p<0.05.
DOI: https://doi.org/10.7554/eLife.46080.027

The following source data is available for figure 4:

**Source data 1.** Activation table for map in *Figure 4*; PPI for value-based decision trials with hippocampus seed modulated by RT.
DOI: https://doi.org/10.7554/eLife.46080.028

p=0.0004). These slower RTs were specific to the value-based compared to the perceptual decision task (p=0.002 for the interaction between task type and group on RT). In addition, their RTs were less driven by subjective value ratings (flatter red curve in *Figure 5B* bottom). This difference between amnesic patients and healthy controls was statistically reliable (p=0.015, interaction between |$\Delta Value$| and group on RT in a mixed effects linear regression, see Materials and methods). In principle, slower decisions could be a sign of a speed-accuracy tradeoff favoring accuracy, but that does not appear to be the case, as the patients were both slower and less accurate (i.e. less consistent with initial subjective values) than the controls. To clarify this point, we calculated an index of efficiency ($I_E$) for each participant (average accuracy divided by the average RT). The index captures the extent to which additional time was used to resolve sources of uncertainty that contribute to stochastic choice behavior. For perceptual decisions, $I_E$ did not differ between amnesic patients

**Table 1.** Amnesic patient demographic and neuropsychological data.

| | | | | | WAIS-III | | WMS-III | | | | | | | |
|---|---|---|---|---|---|---|---|---|---|---|---|---|---|
| Patient # | Diagnosis | Gender | Age | Edu | VIQ | WMI | GM | VD | AD | BNT | FAS | L-N sequence | Years since onset |
| P01 | Hypoxic-ischemic | F | 67 | 12 | 88 | 75 | 52 | 56 | 55 | −1.3 | −1.1 | -2 | 27.29 |
| P02 | Status epilepticus + left temp. lobectomy | M | 54 | 16 | 93 | 94 | 49 | 53 | 52 | −4.6 | −0.96 | -1 | 29.17 |
| P03 | Hypoxic-ischemic | M | 61 | 14 | 106 | 115 | 59 | 72 | 52 | 0.54 | −0.78 | 1.33 | 24.18 |
| P04 | Hypoxic-ischemic | M | 65 | 17 | 131 | 126 | 86 | 78 | 86 | 1.3 | 0.03 | 1.33 | 15.00 |
| P05 | Encephalitis | M | 75 | 13 | 99 | 104 | 49 | 56 | 58 | −0.11 | −0.5 | 0.33 | 5.85 |
| P06 | Stroke | M | 53 | 20 | 111 | 99 | 60 | 65 | 58 | 1.02 | 2.1 | −0.33 | 3.45 |

Age in years at first session; Edu, education in years; WAIS-III, Wechsler Adult Intelligence Scale-III (*Wechsler, 1997a*); WMS-III, Wechsler Memory Scale-III (*Wechsler, 1997b*); VIQ, verbal IQ; WMI, working memory index; GM, general memory; VD, visual delayed; AD, auditory delayed; scores are age-adjusted such that a score of 100 is the age-adjusted mean with a standard deviation of 15; BNT, Boston Naming Test; FAS, verbal fluency test; L-N, Letter-Number Sequence. BNT, FAS and L-N scores were z-scored against normative data for each test.
DOI: https://doi.org/10.7554/eLife.46080.029

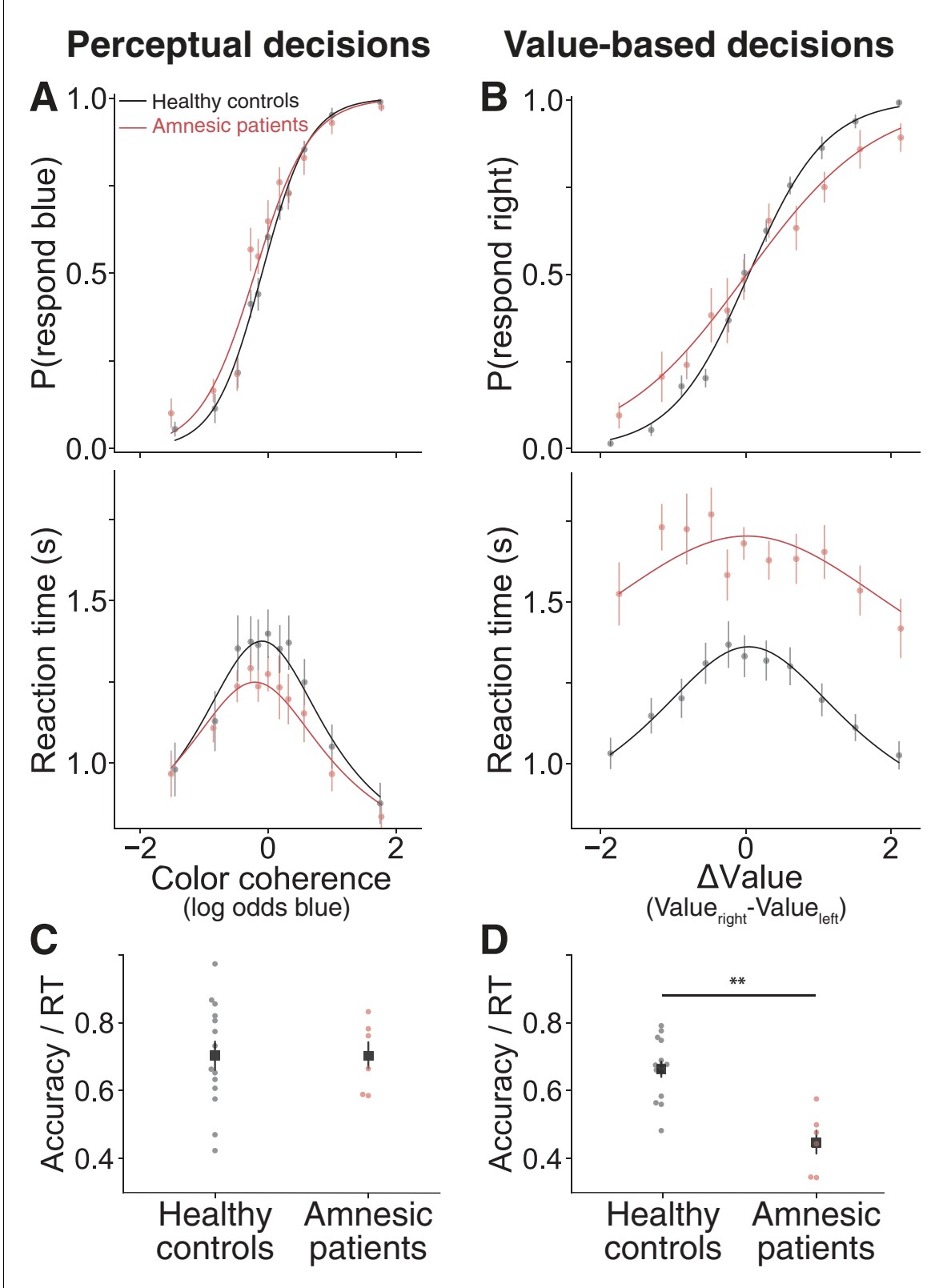

**Figure 5.** Amnesic patients exhibited more stochastic choices and longer reaction times on value-based decisions but not perceptual decisions. (**A**) Proportion of blue choices (top) and mean RT (bottom) plotted as a function of signed color coherence, the logarithm of the odds that a dot is plotted as blue. Data from 14 healthy controls and six amnesic patients (2922 and 1246 trials, respectively). (**B**) Proportion of right-item preference (top) and mean RT (bottom) plotted as a function of value difference (right minus left) binned into 11 levels. Data from 14 healthy controls and six amnesic

*Figure 5 continued on next page*

*Figure 5 continued*

patients (2893 and 1118 trials, respectively). To further summarize these findings, we plot individual average speed-adjusted accuracy, calculated as average accuracy divided by average RT per participant during (C) perceptual decisions and (D) value-based decisions (here, accuracy is defined as choices that are consistent with the individuals' initial value ratings). Circle symbols are data from amnesic patients (red) and healthy age-matched controls (black). Square symbols are group averages. Error bars are s.e.m. Curves are fits of a bounded drift diffusion model (see Materials and methods). See *Figure 5—figure supplement 4* for fits to data from individual participants, *Figure 5—source data 1* for model parameters fit to data from individual participants, and *Figure 5—figure supplement 2* for consideration of an alternative model.

DOI: https://doi.org/10.7554/eLife.46080.030

The following source data, source code and figure supplements are available for figure 5:

**Source code 1.** Jupyter notebook with analysis code and output for analyses performed on data from Experiment 2.
DOI: https://doi.org/10.7554/eLife.46080.035

**Source data 1.** Parameter estimates and goodness of fit measures for Experiment 2.
DOI: https://doi.org/10.7554/eLife.46080.036

**Source data 2.** Trial-level data for the perceptual task in Experiment 2.
DOI: https://doi.org/10.7554/eLife.46080.037

**Source data 3.** Trial-level data for the value-based task in Experiment 2.
DOI: https://doi.org/10.7554/eLife.46080.038

**Figure supplement 1.** Distributions of value ratings and resulting ΔValues used during the choice phase.
DOI: https://doi.org/10.7554/eLife.46080.033

**Figure supplement 2.** Support for a qualitative prediction of a heuristic decision strategy in the amnesic patient group.
DOI: https://doi.org/10.7554/eLife.46080.034

**Figure supplement 3.** Brain images for five out of six amnesic patients included in experiment 2.
DOI: https://doi.org/10.7554/eLife.46080.032

**Figure supplement 4.** Data and fits for value-based and perceptual decisions per participant in Experiment 2.
DOI: https://doi.org/10.7554/eLife.46080.031

and healthy controls (*Figure 5C*, $t_{17.21}$ = 0.02, p=0.98, Welch's t-test), presumably because the uncertainty originates in the stimulus and its noisy representation by sensory neurons (*Britten et al., 1993*; *Mante et al., 2013*; *Shadlen and Newsome, 1998*). For value-based decisions, $I_E$ was significantly lower in the amnesic patients compared to controls (*Figure 5D*, $t_{11.84}$ = 4.2, p=0.0007, Welch's t-test). This implies that whatever deliberative process the amnesics engaged in to reach their decisions, it was less efficient than the process used by the controls.

To further characterize differences in the deliberative process between the groups, we evaluated an alternative to the drift-diffusion model. In this 'heuristic model', the decision maker makes (1) fast choices for items they like strongly, (2) fast choices for an item paired with one they dislike strongly, and (3) slow stochastic choices when the preference is not resolved by rules 1 and 2 (see Materials and methods and *Figure 5—figure supplement 2*). The model is representative of a class of alternatives that would account for RT and choice based on distinct rules—that is, a break from sequential sampling with optional stopping. While we found no support for this model in healthy controls (DDM performs better than this heuristic model, BIC = 537.5), at least one feature of the RTs from the amnesic patients is consistent with this model (*Figure 5—figure supplement 2*). This observation does not provide definitive support for the heuristic above, but it does suggest that the measurable differences between amnesics and controls in accuracy and RT may be related to a fundamental difference in how the amnesics resolve value-based preferences.

## Discussion

We found converging evidence from fMRI and patients pointing to a role for the hippocampus in deliberation between choice options in value-based decisions. In healthy participants, the time it took to resolve choices between two options was longer for near-value decisions and was correlated with BOLD activity in the hippocampus. Amnesic patients with damage to the hippocampus were just as fast as healthy controls to make perceptual decisions but took almost twice as much time to make value-based decisions. The additional time did not lead to better accuracy; in fact, the patients' choices were less accurate (i.e. more stochastic, relative to the values they initially assigned to the items). Together, these findings link the timing of value-based decisions about highly familiar options to the hippocampus.

Value-based decisions between highly familiar choice options are typically assumed to rely on subjective value (*Levy and Glimcher, 2012*; *Rangel et al., 2008*; *Tversky and Kahneman, 1986*). Such value signals are thought to be supported by the ventromedial prefrontal cortex (vmPFC, *Camille et al., 2011*; *Fellows, 2016*; *Fellows and Farah, 2007*; *Levy and Glimcher, 2011*; *Padoa-Schioppa and Assad, 2006*). Yet, even when choosing between options that differ greatly in their subjective value, such choices involve a comparison of the values by way of taking both options, their relation, and their predicted value, into account (*Houston et al., 1999*; *Tversky, 1972*; *Voigt et al., 2017*). Resolving the choice between two options with similar value likely requires the generation of additional information—that is, evidence—to resolve the indecision. This evidence must come from internal sources and might involve multiple dimensions of comparisons between the options. In that sense, it may seem obvious that deliberating between even highly familiar options is likely to involve the sort of relational mechanisms that the hippocampus is known to support.

Our findings suggest that the role of the hippocampus in value-based decisions is almost certainly more nuanced than memory retrieval of the value associated with each of the items. Prior work suggests that simple object-value associations do not depend on the hippocampus (*Neubert et al., 2015*; *Reynolds et al., 2001*; *Rudebeck et al., 2008*; *Rushworth et al., 2011*; *Schultz et al., 1997*; *Vo et al., 2014*). Moreover, it is not obvious why a simple associative memory process would account for longer deliberation times. Instead, we propose that the hippocampus contributes to deliberative processes during decision making. Specifically, we propose that deliberation may be served by the construction of value from internal evidence and engagement in the comparison between the options. Such a process is likely to also involve evaluation of alternatives and prospection about future hypothetical experiences. Prior work suggests that all these processes are likely to engage the hippocampus (*Barron et al., 2013*; *Eichenbaum and Cohen, 2001*; *Schacter et al., 2007*). Future work will be necessary to evaluate how these different processes interact and whether their unique contributions may differ under different circumstances.

Our findings extend recent results demonstrating a role for the hippocampus in value-based decisions under conditions in which value information has been experimentally manipulated to depend on retrieval of new associative memories (*Barron et al., 2013*; *Gluth et al., 2015*; *Wimmer and Shohamy, 2012*). Recent work has also characterized sampling processes during value-based decisions that are reliant on memory (*Bornstein and Norman, 2017*; *Bornstein et al., 2017*; *Duncan and Shohamy, 2016*). Our study builds on these findings to implicate the hippocampus functionally and establish a causal role for the hippocampus in decisions about familiar options for which value is known. One open question is whether this role varies as a function of the nature of the items under deliberation. For example, natural versus packaged items may vary in the extent to which perceptual features reveal their value; the color of an apple is revealing of its sweetness, the color of a package of chocolate perhaps less so. But ultimately, all such decisions depend on the transformation of external perceptual input to internal estimates of subjective value bearing on the relative desirability of the items. It is this deliberative process—beyond the simple item-value association—that we posit the hippocampus contributes to.

The pattern of behavior among the amnesic patients provides further insight into how and when the hippocampus is necessary for value-based decision making. We found that amnesic patients were somewhat less consistent in their decisions and that they took much longer to make them. A similar pattern has been shown recently in healthy older adults with mild memory deficits (*Levin et al., 2018*). As noted earlier, it is unlikely that amnesic patients simply cannot remember the value of the items, as their choices are not arbitrary. This suggests that the patients may be relying on degraded value signals that are coarser than those in controls. Studies of simple valuation have described general valence signals in neurons in orbitofrontal cortex, striatum, amygdala, and anterior cingulate cortex that could potentially drive these choices (*Figure 3—figure supplement 5*; *Hayden et al., 2009*; *Hikosaka et al., 2014*; *Padoa-Schioppa and Assad, 2006*; *Platt and Plassmann, 2014*; *Saez et al., 2017*). Interestingly, patients with vmPFC damage also show greater stochasticity in their choices (*Camille et al., 2011*; *Fellows and Farah, 2007*; *Pelletier and Fellows, 2019*), but do not display the slowing in RT during deliberation that we see in the patients with amnesia due to hippocampal damage. This finding and others (*Jones and Wilson, 2005*; *Wikenheiser et al., 2017*; *Wimmer and Büchel, 2016*), point to possible complementary roles for the hippocampus and the vmPFC in guiding value-based decisions (also see, *McCormick et al.,*

*2018*), with the hippocampus possibly supporting evidence-based construction of value and deliberation (*Weilbächer and Gluth, 2016*).

If patients resolve their choices by accessing a simpler form of value representation, then why do they take such a long time to reach decisions? We propose that this reflects the patients' attempt to engage hippocampal relational mechanisms and their failure to do so. This conclusion is based on a detailed consideration of the relationship between time, accuracy, and choices. In particular, it may help to elaborate on an important difference between the decision processes at play in the value and perceptual decisions we studied. For both tasks, choice and RT were reconciled by fits to drift diffusion models, indicating that both perceptual and value-based decisions exhibit a systematic relationship in speed and accuracy as a function of difficulty. In the perceptual task, a sequence of samples of blue and yellow dots can be converted by the visual system to samples of evidence by *spatially integrating* blue or yellow (or the difference) across the stimulus aperture in sampling epochs governed by the temporal resolution of the color system, which is slower than the frame rate of the display. These samples arrive in series until the subject terminates the decision. The samples are independent, identically distributed random values drawn from a distribution with an expectation (i.e. mean) determined by the stimulus strength and a variance governed by the stochastic properties of the stimulus and the neurons that represent blue, yellow or blue minus yellow. The accumulation of these noisy samples is analogous to a deterministic drift plus diffusion.

As mentioned earlier, similar logic has been applied to value-based decisions (*Krajbich and Rangel, 2011*; *Milosavljevic et al., 2010*; *Polanía et al., 2015*), but the analogy breaks down at the nature of the evidence samples. One might posit that neurons that represent value provide the samples of evidence (*Rangel et al., 2008*; *Rangel and Clithero, 2014*; *Sokol-Hessner et al., 2012*). However, the stimulus provides only one sample of the objects, and there is no reason to think that the brain would then generate a sequence of independent samples of ΔValue (*Shadlen and Shohamy, 2016*). Instead, we reason that the comparison itself triggers constructive thought processes to provide samples of evidence that bear on evaluation of the items along a dimension. It is hard to imagine integrating these samples of ΔValue along different dimensions, although it is possible if they were converted to some common currency (e.g. *Kira et al., 2015*). It seems at least equally likely that each sample leads to a new internal estimate of preference, only to terminate if such a sample provides a sufficiently compelling preference. Although such a process involves no integration, the drift diffusion model can be fit to such a process well enough to render these alternatives indistinguishable (*Ditterich, 2006*). On this view, the longer RTs in the amnesic patients stem from their continued effort to generate evidence to resolve the comparison. Accordingly, the greater stochasticity in their choices possibly stems from the fact that they may fail to generate such evidence and ultimately fall back on a more rudimentary and noisier form of value representation to guide their choices. We are not committed to this specific interpretation and consider a simple heuristic strategy that accounts for some aspects of the data (see *Figure 5—figure supplement 2*).

One limitation of the present study is that we are unable to identify the specific hippocampal-based process that guides deliberation. We can only observe the manifestation of the process in RT and its associated changes in hippocampal BOLD activity or the effect of hippocampal damage. In future work, it will be useful to guide the dimensions of inquiry (e.g. saltiness) and/or construct memories associated with these dimensions that have discernible effects on BOLD activity. In this study, we deliberately avoided any possibility of biasing participants to adopt a memory-based strategy to resolve value preference, as we were interested in testing whether memory spontaneously contributed to such decisions without instruction or guidance.

The idea that memory supports construction of evidence to guide value-based decisions offers new insights to our understanding of how decisions are made, as well as the role of the hippocampus in guiding behavior. The finding that the hippocampus supports deliberation between choice options with similar subjective value addresses a challenge that has long puzzled economists and philosophers (often referred to as Buridan's paradox, *Chislenko, 2016*; *Sorensen, 2004*). By linking the hippocampus to choice behavior, this finding also highlights the pervasive and broad role of the hippocampus in guiding actions and decisions. Research on the hippocampus has typically focused on its role in supporting the formation of conscious, declarative memories for episodes of one's life. The current findings add to a growing shift in this point of view, suggesting that the hippocampus may serve a more general purpose in guiding behavior by providing behaviorally relevant input about relational associations to implicitly guide actions and decisions (*Chun and Phelps, 1999*;

*Eichenbaum and Cohen, 2001*; *Hannula et al., 2007*; *Olsen et al., 2016*; *Palombo et al., 2015a*; *Ryan et al., 2000*; *Schapiro et al., 2014*; *Shohamy and Turk-Browne, 2013*; *Wimmer and Shohamy, 2012*).

## Materials and methods

### Human subjects

#### Experiment 1: young healthy fMRI participants

Thirty-three healthy participants were recruited through fliers posted on campus and the surrounding area in New York City. Three participants were excluded from analysis due to excessive motion during MRI scanning. The final sample consisted of n = 30 (19 female), mean age = 24.7 ± 5.5 and self-reported Body Mass Index (BMI) = 23 ± 4.5. No statistical method was employed to pre-determine the sample size. The sample size we chose is similar to that used in previous publications.

All experimental procedures were approved by the Institutional Review Board (IRB) at Columbia University and all scan participants provided signed informed consent before taking part in the study.

#### Experiment 2: amnesic patients and age-matched healthy control participants

Eight patients with amnesia due to damage to the hippocampus and sixteen age-, education- and verbal IQ-matched healthy control participants were recruited to participate in a version of the same study (for details of the differences between the scan study and the patient study, see below). Two patients and two age-matched healthy control participants were excluded; one patient and one healthy participant did not perform the perceptual decision task satisfactorily (i.e. they did not tend to choose the color that was dominant in the stimulus), one healthy participant did not perform the value-based decision task satisfactorily (i.e. their choices were not consistent with their initial preference ratings) and one patient never completed the perceptual decision task. The final sample included n = 6 patients (one female) with amnesia (see *Table 1* for demographic and neuropsychological data) and n = 14 (6 female) healthy controls matched for age (61.6 ± 10.5), education (15.7 ± 3.6), and verbal IQ (WAIS-III VIQ = 109.5 ± 10.2). All patients presented with severe anterograde and retrograde amnesia. Patients had lower than normal memory scores (two to three standard deviations below normal as measured by WMS-III, *Table 1*), but were largely within normal range for measures of working memory and verbal aptitude. Lesions of five of the MTL patients are presented in *Figure 5—figure supplement 3*, either on MRI or CT images. The remaining patient (P04) had suffered cardiac arrest and could not be scanned due to medical contraindications. MTL pathology for this patient was inferred based on etiology and neuropsychological profile. For the patient who suffered encephalitis (P05), clinical MRI was acquired only in the acute phase of illness, with no visible lesions observed on T1-weighted images. However, T2-flair images demonstrated bilateral hyperintensities in the hippocampus and MTL cortices, as well as the anterior insula. Within the MTL, two patients (P03, P06) had lesions restricted to the hippocampus, while three patients had volume loss extending outside of the hippocampus (P01, P02, P05). For four of the patients (P02, P03, P05, P06), it was possible to determine that the lesion overlapped with the peak of hippocampal activation in the fMRI study. All patients and age-matched healthy participants provided informed consent in accordance with the Institutional Review Boards at Boston University and the VA Boston Healthcare System.

### Tasks

#### Experiment 1

The study took place over two sessions. On the first day, participants were not scanned. They were trained on the perceptual color dots task (details below), received feedback (correct/incorrect) on each trial during training, and were trained to criterion, defined as 80% or higher accuracy over the last four blocks of 10 trials. Training consisted of a minimum 200 trials and a maximum 400 trials. After color dots training, participants underwent incidental encoding for the Memory Localizer task: they rated 100 neutral objects, presented one at a time on the computer screen, on how much they

liked that object by placing the cursor along a visual analog scale that ranged from 0 (least) to 10 (most) using the computer mouse. This liking rating task served as a memory encoding phase, followed two days later by a surprise memory recognition test in the scanner (details below). The first study session lasted about 1 hr. When it ended, participants were told to refrain from eating or drinking anything besides water for four hours before their next appointment. On the second session, which took place 2 days after the first session, participants took part in an auction outside of the scanner. They were then placed in the MRI scanner and performed the food choice task, the color dots task, and the memory recognition task.

## Auction

Participants were endowed with $3, which they used to take part in an auction. The auction followed Becker-Degroot-Marschak (BDM) rules (*Becker et al., 1964*). This auction procedure allowed us to obtain a measure of willingness-to-pay (WTP) for each of 60 appetitive food items per participant (*Plassmann et al., 2007*). Participants were presented with one snack item at a time, in random order, on a computer screen. They placed their bid by moving a cursor on an analog scale that spanned from $0 to $3 at the bottom of the screen using the computer mouse. The auction was self-paced, and the next item was presented only after the participant placed his or her bid. After participants placed bids on all 60 items, they were given a chance to revise their bids to account for adjustments and scaling effects that can occur after participants experience the full food stimulus set. Participants were presented with each of the 60 items in random order a second time with their original bid displayed below and were asked whether they wanted to change their bid. If they clicked 'NO', they were presented with the next food item, and their original bid was kept as the WTP for that item. If participants clicked 'YES', the $0 to $3 analog scale was presented and they placed a new bid using the mouse as before. This new bid was recorded as the final WTP for that item. The starting location of the cursor along the analog scale was randomized on each trial and the mouse cursor was reset to the middle of the screen on each trial to prevent participants from simply clicking through the entire auction phase without deliberation. Participants were told that a single trial would be drawn at random at the end of the session, and that they could bid any amount of the full $3 for each food item because the auction repeated in an independent fashion for each of the 60 items. They were told that their best strategy to win the auction was to bid exactly what each item was worth to them to purchase from the experimenter at the end of the experiment and that bidding consistently high or consistently low was a bad strategy. At the end of the session, the computer generated a counter bid in the form of a random number between $0 and $3 in increments of 25 cents. If the computer bid was higher than the participant's bid, then he or she lost the auction; if the participant matched or outbid the computer, he or she was offered to purchase the randomly drawn food item from the experimenter at the lower price generated by the computer. The outcome of the auction was played out at the end of the experimental session. After performing the auction outside the scanner, participants performed the following three tasks in the scanner while functional brain images were acquired.

## Food choice

The 60 food items were rank-ordered based on WTPs obtained during the auction, and 150 unique pairs made up from the 60 items were formed such that the difference in WTP between the two items in a pair (i.e. ΔValue) varied. Each of the 60 items appeared in five different pairs. Pairs were presented in random order, one pair at a time, with one item on each side of a central fixation cross. Right/left placement of the higher-value item in a pair was counterbalanced across trials. Participants were instructed to choose the food they preferred. Participants chose one item on each trial by pressing one of two buttons on an MRI-compatible button box. They were given up to 3 s to make their selections. After a choice was made, the selected item was highlighted for 500 ms. If participants did not make a choice before the 3 s cutoff, the message 'Please respond faster' was displayed for 500 ms. Trials were separated by a jittered inter-trial-interval (ITI) drawn from an exponential distribution with a mean of 3, if the value generated was below 1 or above 12, it was redrawn. The true average of the resulting distribution of ITIs across trials was 3.05 s with an sd = 2.0 s. Participants were told that they would be given the chosen food on a single randomly selected trial to eat. Participants were presented with 210 trials total, split into three 70-trial scan

runs. Runs of the food choice task were interleaved with runs of the color dots task (below). Of the 150 unique pairs, 90 pairs were presented only once and 60 pairs were presented twice. Thus, each of the 60 food items was presented 7 times in total. Each scan run of the food choice task lasted 7 min.

## Color dots

Participants viewed a dynamic random dot display and were asked to determine whether there were more yellow or blue dots. Dots were presented at random locations within a central circular aperture (diameter 5 cm) and replaced in each video frame (60 Hz) by new dots (density 16.7 dots•cm$^{-2}$•s$^{-1}$) at new random positions. Each dot was assigned a color randomly with probability controlled by the color coherence, $C = log(p_{blue}/p_{yellow})$, such that $p_{blue} = \frac{e^C}{1+e^C}$ and $p_{yellow} = 1 - p_{blue}$. A dot that is not blue is yellow. Throughout a single trial, $C$ was fixed at a value drawn from a set of 11 possible levels {−2, −1, −0.5, −0.25, −0.125, 0, 0.125, 0.25, 0.5, 1, 2}. For $C$>0 ($p_{blue}$>0.5) a blue choice is graded as correct regardless of the actual ratio of blue:yellow dots displayed. For $C$<0 ($p_{yellow}$>0.5) a yellow choice is graded as correct regardless of the actual ratio of blue:yellow dots actually displayed. For $C = 0$ ($p_{blue} = P_{yellow} = 0.5$), the assignment of correct was deemed 0.5. The color strength is $|C|$.

Participants responded by pressing one of two buttons, with the color-button mapping counterbalanced across participants. Participants were instructed to make their response as soon as they had an answer. The stimulus was presented for a maximum of 2.5 s. If they responded within the 2.5 s window, the stimulus disappeared and a central fixation cross reappeared. Intertrial intervals were generated using the same procedure used for the value-based decision task, resulting in a distribution mean across trials of 3.04 s with an sd = 2.4. Participants did not receive feedback during the main data collection, but on session 1 (training; no scanning) they received correct/incorrect feedback on each trial. Feedback appeared after a response was made and remained on the screen for 500 ms. If they did not respond within the 2.5 s choice window, a message asking participants to please respond faster was displayed for 500 ms. Participants were presented with a total of 210 trials, split into three scan runs of 70 trials each. Each scan run of the color dots task lasted 6.5 min.

## Memory recognition

Participants were presented with the 100 objects they had rated during session one as well as 100 new objects, randomly intermixed, one object at a time in the middle of the screen. Below the image of the object and to the right and left of center appeared the words 'OLD' and 'NEW' that corresponded to the right/left button mapping. On each trial, participants were asked to determine whether the object on the screen was old, meaning they remembered rating that object on their first visit or if the object was new, meaning they did not remember seeing or rating that object. Participants responded by pressing one of two buttons on an MRI-compatible button box. Old/New response-button mapping was counterbalanced across participants. The stimulus remained on the screen for a maximum of 3 s. If participants responded within the 3 s response window, their choice (i.e. OLD or NEW) was highlighted for 500 ms. If they did not respond within the 3 s window, a message asking them to please respond faster was displayed for 500 ms. Trials were separated by a jittered ITI generated using the same procedure as for the other two tasks and resulted in a distribution mean across trials of 3.0 s with an sd = 1.98 s. The 200 trials were split into four scan runs of 50 trials (approximately 5 min) each. All four runs of this task were consecutive, with no other intervening tasks in between.

## Experiment 2

The patients and age-matched healthy participants performed a version of the scan study that did not include the memory recognition task and was performed outside of the scanner. The study was conducted over two days; on one day participants took part in the value-based decision task and on the other day they took part in the perceptual decision task. The order in which tasks were performed was counterbalanced across participants. The value-based decision task differed from the task in Experiment 1 in four ways. (1) The food stimuli used were different and consisted of a wider range of non-packaged foods, not just snack foods. (2) Rather than a BDM auction, participants indicated their pre-experimental preferences for 30 food items using a preference rating scale. Participants were instructed to rate how much they prefer to eat the food item on the screen from 0

(prefer least to eat) to 10 (prefer most to eat). Participants were asked to name the food item on the screen before rating it. Foods that a participant did not recognize or misnamed were excluded from analysis. This ensured that only familiar foods were included in the analysis. Ratings were z-scored within participant and ΔValue was calculated from the z-scored ratings for 210 unique pairs of items, none of which repeated during the food choice task. (3) Participants were given 3.5 s rather than 2.5 s to make a choice. (4) Participants were not asked to fast before the experiment and did not receive a snack at the end of the experiment based on their choice on a randomly selected trial. The perceptual decision task differed from the task in Experiment 1 in three ways: (1) participants received only 40 practice trials, (2) participants continued to receive correct/incorrect feedback throughout the entire task, and (3) participants were given 3.5 s to make a choice. Prior to the perceptual task, participants were trained on selecting blue or yellow using the proper button on the keyboard to ensure that they learned the color-button mapping prior to starting the perceptual decision task. Participants were trained for only 40 trials rather than to criterion and continued to receive correct/incorrect feedback for all trials.

## fMRI acquisition

Imaging data were acquired on a 3 T GE MR750 MRI scanner with a 32-channel head coil. Functional data were acquired using a T2*-weighted echo planar imaging sequence (repetition time (TR) = 2 s, echo time (TE) = 22 ms, flip angle (FA) = 70˚, field of view (FOV) = 192 mm, acquisition matrix of 96 × 96). Forty oblique axial slices were acquired with a 2 mm in-plane resolution positioned along the anterior commissure-posterior commissure line and spaced 3 mm to achieve full brain coverage. Slices were acquired in an interleaved fashion. Each of the food choice runs consisted of 212 volumes, each of the color dots runs consisted of 197 volumes, and the memory test runs consisted of 150 volumes. In addition to functional data, a single three-dimensional high-resolution (1 mm isotropic) T1-weighted full-brain image was acquired using a BRAVO pulse sequence for brain masking and image registration.

## Behavioral analysis

### Choice and reaction time data

Choice and RT data were analyzed using regression models. Choice data were scored on accuracy (correct choice in the perceptual decision task or consistency of responses with the stated value for the choice option—WTP for the scan study and preference rating for patient study—that is score one for trials when the participant chose the food with higher WTP/rating and 0 if they chose the food with lower WTP/rating). These binary data were then entered into a repeated measures logistic regression mixed effects model to calculate the odds of choosing correctly/consistently with their prior valuation and test the relationship between choices and task difficulty (color coherence or ΔValue). ΔValue for the scan study was calculated by subtracting the WTP for the item on the left side of the screen from the WTP for the item on the right side of the screen. ΔValue for the patient study was calculated by subtracting the z-scored rating (z-scored within participant) for the item on the left from the z-scored rating of the item on the right. RT data were entered into a mixed effects repeated measures linear regression model to test the relationship between RT and |color coherence| or |ΔValue|. For the patient study, we also entered group assignment as a predictor in the models and its interaction with ΔValue separately for each task. For the patient study, we also ran a full model combining across both tasks to assess the three-way interaction between group (patient or healthy), task type (perceptual or value-based), and difficulty on choices or RT.

### Drift diffusion model

We fit a one-dimensional drift diffusion model to the choice and RT on each decision. The model assumes that choice and RT are linked by a common mechanism of evidence accumulation, which stops when a decision variable reaches one of two bounds. The decision variable ($x$) is given by the cumulative sum of samples from a Normal distribution with mean $\mu dt$ and variance $dt$,

$$dx = \mu dt + \mathcal{N}(0, dt) \tag{1}$$

where $\mathcal{N}$ represents an independent sample from a Normal distribution with mean 0 and variance $dt$, that is, the increment of a Wiener process. The accumulation starts with $x = 0$.

In the value-based decision, the mean of the momentary evidence (also termed the drift rate) is given by

$$\mu = \kappa s |V_R - V_L|^p + \mu_0 \qquad (2)$$

where $V_R$ and $V_L$ are the values of the right and left item respectively, $\kappa$ is a fitted constant, $s$ is the sign of the value difference (positive if $V_R > V_L$, negative otherwise), $p$ is a fitted exponent and $\mu_0$ implements a bias to the drift rate to account for non-symmetric distributions of choice or RT between left and right choices. If $p = 1$, then $\kappa s$ would yield a drift rate that varies linearly as a function of $\Delta Value$. The power law instantiates the possibility that the monotonic relationship between $\Delta Value$ and drift rate is not necessarily linear. $\kappa$, $p$, and $\mu_0$ are fitted parameters.

In the color-discrimination task, the mean of the momentary evidence is given by

$$\mu = \kappa s |C|^p + \mu_0 \qquad (3)$$

where $C$ is the color coherence, and $s$ is positive if $C > 0$ and negative otherwise. There is reason to expect $p \approx 1$ for the color-discrimination task, but we allowed this degree of freedom (for parity).

We used time-varying decision bounds to account for potential differences in RT between correct and error trials. This is the normative implementation of bounded drift diffusion when there are multiple difficulty levels (*Drugowitsch et al., 2012*). The shape of the bound was determined by three parameters. The initial bound height, $B_0$, remains constant for $0 \leq t < B_{del}$, and then collapses exponentially towards zero with time constant $B_2$ (in seconds). The two bounds were assumed to be symmetrical around $x = 0$. For the value-based task, the positive bound represents a commitment to a right-item choice, and the negative bound represents a commitment to a left item choice. For the perceptual task, the positive and negative bounds indicate a commitment to the blue and yellow choices, respectively. The RT is given by the sum of the decision time, determined by the drift-diffusion process, and a non-decision time that we assume Gaussian with mean $t_{nd}$ and standard deviation $\sigma_{tnd}$.

We performed separate fits for perceptual- and value-based tasks. The model was fit to maximize the joint likelihood of choice and RT of each trial. The likelihood of the parameters given the data from each trial was obtained by numerically solving the Fokker-Planck (FP) equation describing the dynamics of the drift-diffusion process. We used fast convolution methods to find the numerical solution to the FP equation. The parameter optimization was performed using the Nelder-Mead Simplex Method (*Lagarias et al., 1998*) to minimize the negative log-likelihood of the parameters given the choice and RT data. All parameters were bounded during the fitting procedure. We took the best fit parameters from one hundred fits using random starting points to ensure that the optimization search did not get stuck in a local minimum. For the value-based task, we reduced the number of unique drift-rates by rounding $\Delta Value$ to multiples of 0.1 dollars. In *Figure 2* and *Figure 5*, we fit the models to grouped data from all participants after binning $\Delta Value$ into 11 levels. These 11 levels had fixed boundaries on $\Delta Value$ and were assigned the mean $\Delta Value$ of the points composing the bin. This binning was intended to match the levels of $\Delta Value$ to the discrete levels of color coherence. The fits for individual participants were performed on all trials (not binned) and are shown in *Figure 2—figure supplement 1* and *Figure 5—figure supplement 4*. The best fitting parameters for the grouped and non-grouped data are displayed in *Figure 2—source data 1* and *Figure 5—source data 1*.

## Heuristic model

We evaluated an alternative to drift-diffusion models, which obeys the following heuristic. Suppose that a subset of food items are valued as either highly desirable ($D^+$) or highly undesirable ($D^-$). All the other items are designated middling ($D^\approx$). This yields three types of decisions: (i) Decisions between an item from $D^+$ and an item from the other categories ($D^-$ or $D^\approx$) are fast choices of the preferred item regardless of $\Delta Value$. (ii) Decisions between an item from $D^-$ and an item from $D^\approx$ are fast choices of item from $D^\approx$ regardless of $\Delta Value$. (iii) Decisions between two items from the same class (both from $D^+$, both from $D^-$, or both from $D^\approx$) are slow, regardless of $\Delta Value$ and they are stochastic. We allowed these stochastic choices to be governed by a logistic function of $\Delta Value$, although it could be argued that they ought to be random. We refer to i and ii as *trivial* decisions and to iii as *non-trivial* decisions. The only role of $\Delta Value$ is to determine the choice probabilities for

the *non-trivial* decisions. Importantly, it is uncoupled to the RT, which is uniformly slow for this category.

We implemented this model using the following degrees of freedom: $\kappa_1$ and $\kappa_2$ are criteria that separate the ranges of value corresponding to $D^-$, $D^\approx$ and $D^+$; two means and two standard deviations for the fast and slow RTs; and two degrees of freedom ($\beta_0$, $\beta_1$) for the logistic regression relating the *non-trivial* choices to $\Delta Value$. The model was fit to maximize the joint likelihood of choice and RT of each trial. We used the Nelder-Mead Simplex Method (*Lagarias et al., 1998*) to find the model's parameters that minimize the negative log-likelihood (NLL) of the choice and RT data. $RT_i$ and $RT_{ii}$ are assumed to be generated from a normal distribution with a mean $\mu_{RTfast}$ and a standard deviation $\sigma^2_{RTfast}$. $RT_{iii}$ are assumed to be generated from $\mu_{RTslow}$ and a standard deviation $\sigma^2_{RTslow}$. The NLL for *non-trivial* choices derive from a Bernoulli (binomial) distribution: $-log(p[choice|,\beta_0,\beta_1])$. The NLL for *trivial* choices is not properly specified. The model posits a deterministic decision rule for these trials, but the data exhibit stochasticity (see insets in *Figure 5—figure supplement 2*). To avoid infinite penalization during fitting, we assigned the probability $p = 0.99$ for *trivial* choices consistent with the rule, and $1 - p$ for the exceptions. For model comparison statistics (e.g. BIC), we obtain p from the logistic function (derived from the *non-trivial* choices) evaluated at Max($|\Delta Value|$).

The arbitrary choice of penalty for inconsistent choices on *trivial* trials renders model comparison ill-posed. The same can be said for the implementation of a logistic choice function to account for the stochastic *non-trivial* choices. Nevertheless, we compared the heuristic model to the diffusion models by comparing the deviance of the best fits (same as BIC because the number of degrees of freedom are equal). We also implemented a version of the model that employs a 'trembling hand' error for penalizing an incorrect choice on a *trivial* trial by allowing the probability p for *trivial* choices to be a free parameter. We find that the DDM model still performs better than this more permissive parametrization of the heuristic model (BIC = 425.88).

The unsatisfactory aspects of this model comparison exercise led us to pursue a more qualitative strategy. The heuristic model posits independence of RT and $\Delta Value$ once grouped by *trivial* or *non-trivial*, whereas sequential sampling models (e.g. diffusion) predict a dependence regardless of this grouping. We evaluated this prediction by examining the effect of $|\Delta Value|$ on RT, using mixed effects linear regression on combined data from the participants in the three experimental groups: imaging, amnesic patients and their age-matched controls. For the heuristic model, the designation of *trivial* vs. *non-trivial* was established from fits to each participant's data (i.e., $\kappa_1$ and $\kappa_2$). The analysis is shown in *Figure 5—figure supplement 2*.

## Imaging analysis

### Imaging data preprocessing

Raw imaging data in DICOM format were converted to NIFTI format and preprocessed through a standard preprocessing pipeline using the FSL package version 5 (*Smith et al., 2004*). Functional image time series were first aligned using the MCFLIRT tool to obtain six motion parameters that correspond to the x, y, z translation and rotation of the brain over time. Next, the skull was removed from the T2* images using the brain extraction tool (BET) and from the high-resolution T1 images using Freesurfer (*Fischl et al., 1999*; *Ségonne et al., 2004*). Spatial smoothing was performed using a Gaussian kernel with a full-width half maximum (FWHM) of 5 mm. Data and design matrix were high-pass filtered using a Gaussian-weighted least-squares straight line fit with a cutoff period of 100 s. Grand-mean intensity normalization of each run's entire four-dimensional data set by a single multiplicative factor was performed. The functional volumes for each participant and run were registered to the high resolution T1-weighted structural volume using a boundary-based registration method implemented in FSL5 (BBR, *Greve and Fischl, 2009*). The T1-weighted image was then registered to the MNI152 2 mm template using a linear registration implemented in FLIRT (12 degrees of freedom). These two registration steps were concatenated to obtain a functional-to-standard space registration matrix.

### Food choice

We conducted a generalized linear model (GLM) analysis on the food choice task data. The first analysis included three regressors of interest: (i) onsets for all valid choice trials; (ii) same onsets and duration as (i) but modulated by RT; (iii) onsets for missed trials. After running this model, we ran a

conjunction analysis using the output of this model and the equivalent model on the perceptual decision task data (see below) with our main memory retrieval success contrast (see memory recognition section below). The conjunction map is presented in *Figure 3*. This model was also used to generate the map in *Figure 3—figure supplement 2A*.

The second GLM analysis was designed to rule out the possibility that differences in RT range between the two tasks might account for a contrast between tasks in the effect of RT on BOLD. This model included five regressors of interest: (i) onsets for all valid choice trials with RT in the range of overlap across the two tasks; (ii) same onsets and duration as (i) but modulated by RT; (iii) onsets for all valid choice trials with RT not in the range of overlap across the two tasks; (iv) onsets for missed trials. This model was used to generate the map in *Figure 3—figure supplement 2B*.

The third GLM model is the model we based our inferences on and included twelve regressors of interest: (i) onsets for non-repeated unique pair 'correct' trials (i.e. unique pairs of items that were only presented once where choice was consistent with initial valuation during the auction meaning the chosen item had the higher WTP), modeled with a duration that equaled the average RT across all valid food choice trials and participants; (ii) same onsets and duration as (i) but modulated by $|\Delta$ *Value*| demeaned across these trials within each run for each participant; (iii) same onsets and duration as (i) but modulated by RT demeaned across these trials within each run for each participant; (iv-vi) similar to regressors (i-iii), but for non-repeated unique pair 'incorrect' trials (i.e. unique pairs of items that were only presented once for which choice was inconsistent with initial valuation during the auction, meaning the chosen item had the lower WTP); (vii-ix) similar to regressors (i-iii), but for repeated unique pair trials (i.e. unique pairs of items that were presented twice, both 'correct' and 'incorrect' trials together); (x) to account for any differences in mean value across items in a pair (i.e. average WTP across both items in a pair) between trial types, we added a regressor with the onsets of all valid trials and the same duration as all other regressors, while the modulator was the demeaned average WTP across both items in a pair; (xi) to account for any differences in right/left choices between trial types, we added a regressor with the same onsets and durations as (x), while the modulator was an indicator for right/left response; (xii) onsets for missed trials. Maps from this model are presented in *Figure 3—figure supplement 3*.

In all models, we also included the six x, y, z translation and rotation motion parameters obtained from MCFLIRT, framewise displacement (FD) and RMS intensity difference from one volume to the next (*Power et al., 2012*) as confound regressors. We also modeled out volumes with FD and DVARS that exceeded a threshold of 0.5 by adding a single time point regressor for each 'to-be-scrubbed' volume (*Siegel et al., 2014*). All regressors were entered at the first level of analysis and all (but the added confound regressors) were convolved with a canonical double-gamma hemodynamic response function. The temporal derivative of each regressor (except for the added confound regressors) was included in the model. The models were estimated separately for each participant and each run.

## Color dots

The first GLM analysis included three regressors of interest: (i) onsets for all valid choice trials; (ii) same onsets and duration as (i) but modulated by RT; (iii) onsets for missed trials. After running this model, we ran a conjunction analysis using the output of this model and the equivalent model on the value-based decision task data (see above) with our main memory retrieval success contrast (see memory recognition section below). The conjunction map is presented in *Figure 3*. This model was also used to generate the map in *Figure 3—figure supplement 2A*.

The second GLM analysis evaluates the possibility that differences in RT variance between the two tasks might account for a contrast between tasks in the effect of RT on BOLD. This model included five regressors of interest: (i) onsets for all valid choice trials with RT in the range of overlap across the two tasks; (ii) same onsets and duration as (i) but modulated by RT; (iii) onsets for all valid choice trials with RT not in the range of overlap across the two tasks; (iv) onsets for missed trials. This model was used to generate the map in *Figure 3—figure supplement 2B*.

The third GLM model for the color dots task is the model that we based our inferences on and included three regressors for each of correct and incorrect color choice trial types: (i) onsets of correct trials (i.e. participant chose yellow when the coherence was negative and chose blue when the coherence was positive, as well as all coherence 0 trials) modeled with a duration which equaled the

average RT across all valid color dots trials and participants; (ii) same onsets and durations as (i) but modulated by |color coherence| demeaned across these trials within each run for each participant; (iii) same onsets and durations as (i) but modulated by RT demeaned across these trials within each run for each participant; (iv-vi) similar to regressors (i-iii), but for incorrect trials (i.e. participant chose yellow when the coherence was positive and chose blue when the coherence was negative). Additionally, we included two other regressors: to account for any differences in right/left choices between trial types we added a regressor (vii) with the onsets of all valid color dots trials and the same duration as all other regressors (average RT across all trials and participants), while the modulator was an indicator for right/left response; finally, we included a regressor (viii) with onsets for missed trials. Maps from this model are presented in *Figure 3—figure supplement 3*.

For all models, we added the same covariates as in the food choice design matrix, including the six motion regressors described above, along with FD and DVARS as confound regressors.

## Memory recognition

The GLM for the memory recognition task data included 8 regressors of interest: (i) onsets of hit trials (i.e. participant responded old when the object on the screen was old), modeled with a duration that equaled the average RT across all valid memory trials and participants; (ii) same onset and duration as (i) but modulated by liking rating for the object demeaned across these trials within each run for each participant; (iii) onsets of miss trials (i.e. participant responded new when the object on the screen was old) modeled with the same duration as (i); (iv) same onset and duration as (iii) but modulated by liking rating for the object demeaned across these trials within each run for each participant; (v) onsets of correct rejection trials (i.e. participant responded new when the object on the screen was new) modeled with the same duration as (i); (vi) onsets of false alarm trials (i.e. participant responded old when the object on the screen was new) modeled with the same duration as (i); (vii) to account for any differences in RT between trial types we added a regressor with the onsets of all valid trials and the same duration as all other regressors (average RT across all trials and participants) while the modulator was the demeaned RT across all valid trials; (viii) onsets for missed trials. We added the same covariates as in the food choice design matrix, including the six motion regressors described above, along with FD and DVARS as confound regressors. The map for the contrast hits >correct rejections in this model is presented in *Figure 3—figure supplement 1*. This contrast was also used in the conjunction analysis presented in *Figure 3*.

## Conjunction analysis

To test the spatial overlap in memory-retrieval-related brain activity and value-based-RT-related activation, we conducted a conjunction analysis between the maps presented in *Figure 3—figure supplement 1* (memory retrieval success contrast of hits [regressor (i) in memory recognition fMRI GLM model] greater than correct rejections [regressor (v) in memory recognition fMRI GLM model]) and the same map as in *Figure 3—figure supplement 3C*, but for the simpler model (contrast of value-based RT [regressor (iii) in the first food choice fMRI GLM model] greater than perceptual RT [regressor (iii) in the first color dots fMRI GLM model]). The conjunction map is presented in *Figure 3*.

## Psychophysiological interaction (PPI)

As the seed for the PPI analysis, we used significant voxels for the contrast value-based RT greater than perceptual RT (*Figure 3—figure supplement 3C*) that lay within an anatomical mask of bilateral hippocampus (Harvard-Oxford Atlas). The PPI regressor was created by deconvolving the seed to obtain an estimated neural signal during value-based decisions using SPM's deconvolution algorithm (*Gitelman et al., 2003*), calculating the interaction with the task in the neural domain and then reconvolving to create the final regressor. We followed McLaren et al.'s (*McLaren et al., 2012*) gPPI modeling procedure and included nine regressors in our GLM: (i) onsets of all valid food choice trials, modeled with a duration that equaled the average RT across all valid trials and participants; (ii) onsets of all valid trials, modeled with the same duration as in i and modulated by RT, demeaned across these trials within each run for each participant; (iii) onsets of all valid trials, modeled with the same duration as in i and modulated by |ΔValue|, demeaned across these trials within each run for each participant; (iv) same onsets and duration as i but modulated by the value of the chosen food, demeaned across these trials within each run for each participant; (v) to account for any differences

in right/left choices, we added a regressor with the same onsets and duration as i but modulated by an indicator for right/left response; (vi) onsets of all missed trials with the same duration as i; (vii) the raw time course extracted from the seed (after registering the seed to the native space of each run for each participant); (viii) a PPI regressor with the same onsets as i. The PPI that varied linearly with RT during food choice trials generated the map in *Figure 4*.

## GLM model estimation and correction for multiple comparisons

All GLM models were estimated using FSL's FEAT. The first-level time-series GLM analysis was performed for each run per participant using FSL's FILM. The first-level contrast images were then combined across runs per participant using fixed effects. The group-level analysis was performed using FSL's mixed effects modeling tool FLAME1 (*Beckmann et al., 2003*). Group-level maps were corrected to control the familywise error rate in one of two ways: for whole-brain correction, we used cluster-based Gaussian random field correction for multiple comparisons, with an uncorrected cluster-forming threshold of $z = 2.3$ and corrected extent threshold of $p < 0.05$. For small volume correction, we used a voxel-based Gaussian random field theory-based maximum height thresholding with a voxel-level corrected threshold of $p < 0.05$ within a 3D mask of a region of interest.

## Data and software availability

Data from this study are available from the corresponding author upon request. Task code and analysis code is available at https://github.com/abakkour/MDMRT_scan (*Bakkour, 2019*; copy archived at https://github.com/elifesciences-publications/MDMRT_scan/settings). Imaging analysis code is available from the corresponding author upon request.

## Acknowledgements

This research was funded by the McKnight Memory and Cognitive Disorders Award (DS), National Science Foundation Directorate for Social, Behavioral and Economic Sciences Postdoctoral Research Fellowship grant #1606916 (AB), National Institutes of Health grant R01EY011378 and Howard Hughes Medical Institute (MNS), VA Senior Research Career Scientist Award (MV), VA Merit Grant CX001748 (MV), National Eye Institute grant T32 EY013933 (YHRK).

The authors would like to thank Daniel Kimmel, Danique Jeurissen, and David Barack for feedback on an earlier draft of the manuscript and helpful discussions. The authors would also like to thank Lucy Owen, Sean Raymond, Eileen Hartnett, and the New York State Psychiatric Institute MR Unit staff for help with data collection. The contents of this manuscript do not represent the view of the US Department of Veterans Affairs or the US Government.

## Additional information

### Funding

| Funder | Grant reference number | Author |
| --- | --- | --- |
| McKnight Foundation | McKnight Memory and Cognitive Disorders Award | Daphna Shohamy |
| National Science Foundation | 1606916 | Akram Bakkour |
| National Institutes of Health | R01EY011378 | Michael N Shadlen |
| Howard Hughes Medical Institute | HHMI Investigator | Michael N Shadlen |
| U.S. Department of Veterans Affairs | VA Senior Research Career Scientist Award | Mieke Verfaellie |
| U.S. Department of Veterans Affairs | VA Merit Grant CX001748 | Mieke Verfaellie |
| National Eye Institute | T32-EY013933 | Yul HR Kang |

The funders had no role in study design, data collection and interpretation, or the decision to submit the work for publication. The contents of this manuscript do not represent the view of the US Department of Veterans Affairs or the US Government.

## Author contributions

Akram Bakkour, Conceptualization, Data curation, Software, Formal analysis, Supervision, Funding acquisition, Validation, Investigation, Visualization, Methodology, Writing—original draft, Project administration, Writing—review and editing; Daniela J Palombo, Supervision, Investigation, Methodology, Writing—review and editing; Ariel Zylberberg, Conceptualization, Software, Formal analysis, Investigation, Visualization, Methodology, Writing—review and editing; Yul HR Kang, Software, Formal analysis, Methodology, Writing—review and editing; Allison Reid, Data curation, Investigation, Project administration; Mieke Verfaellie, Supervision, Methodology, Writing—review and editing; Michael N Shadlen, Daphna Shohamy, Conceptualization, Supervision, Writing—original draft, Writing—review and editing

## Author ORCIDs

Akram Bakkour (iD) https://orcid.org/0000-0002-6070-4945
Ariel Zylberberg (iD) https://orcid.org/0000-0002-2572-4748
Michael N Shadlen (iD) https://orcid.org/0000-0002-2002-2210

## Ethics

Human subjects: Experimental procedures in Experiment 1 were approved by the Institutional Review Board (IRB) at Columbia University through Columbia IRB Protocol #AAAO5907. All fMRI participants provided signed informed consent before taking part in the study. All patients and age-matched healthy participants in experiment 2 provided informed consent in accordance with the Institutional Review Boards at Boston University and the VA Boston Healthcare System outlined in VABHS IRB #2997.

## Decision letter and Author response

Decision letter https://doi.org/10.7554/eLife.46080.044
Author response https://doi.org/10.7554/eLife.46080.045

# Additional files

## Supplementary files

• Transparent reporting form
DOI: https://doi.org/10.7554/eLife.46080.039

## Data availability

Behavioral data from this study are available as source data included in this submission. Behavioral analysis code is available as source code included in this submission. Analysis code as well as task code is available at https://github.com/abakkour/MDMRT_scan (copy archived at https://github.com/elifesciences-publications/MDMRT_scan). Imaging data has been deposited to OpenNeuro and is available to download at https://openneuro.org/datasets/ds002006/versions/1.0.0.

The following dataset was generated:

| Author(s) | Year | Dataset title | Dataset URL | Database and Identifier |
|---|---|---|---|---|
| Akram Bakkour, Daphna Shohamy, Michael N Shadlen | 2019 | Memory and decision making dataset | https://openneuro.org/datasets/ds002006/versions/1.0.0 | OpenNeuro, ds002006 |

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

# Appendix 1

DOI: https://doi.org/10.7554/eLife.46080.040

## Experiment 1 Behavioral Results

### Value-based Decisions

Participants tended to choose the item that was of higher value (as measured by willingness-to-pay in the auction phase), and this tendency increased as the value difference between the two items (i.e. $\Delta Value$) increased (**Figure 2B**, top. The odds of right item choices multiplied for every \$1 increase in $\Delta Value$ by 5.9, 95% CI = [5.08 6.93], p<0.0001). Participants' RTs increased as $|\Delta Value|$ decreased (**Figure 2B**, bottom; $\beta = -0.11$, bootstrapped 95% CI [−0.13–0.09], bootstrapped p=0.001). These results replicate previous findings that show that choices and RTs vary systematically with $\Delta Value$ (**Krajbich et al., 2010**; **Milosavljevic et al., 2010**). These relationships are captured by the drift diffusion model (solid black lines in **Figure 2**), suggesting that the mechanism underlying the decision is based on accumulation of evidence.

### Perceptual Decisions

When performing the color task, participants responded blue more often as the color coherence increased and responded yellow more often as the color coherence decreased (became more negative, **Figure 2A**, top. The odds of choosing blue multiplied for every unit increase of color coherence by 68.05, 95% CI [52.63 87.99], p<0.0001). RTs were shortest at the lowest and highest color coherence levels. RTs were longest at color coherence level zero, when there was an equal proportion of yellow and blue dots in the stimulus (**Figure 2A**, bottom). In a repeated measures linear regression mixed effects model, |color coherence| was negatively related to RT ($\beta = -0.25$, bootstrapped 95% CI [−0.26–0.24], bootstrapped p=0.001).

### Memory recognition

Participants' mean hit rate was $0.81 \pm 0.15$, and their mean correct rejection (CR) rate was $0.76 \pm 0.15$. Mean $d'$ across all participants was $1.78 \pm 0.57$. Participants were faster when making a correct response (hits and CRs combined, mean RT = $1.24 \pm 0.17$) than when making an incorrect response (misses and false alarms combined, mean RT = $1.42 \pm 0.24$, mean of the differences in RT = 0.18, 95% CI [0.13 0.23], t(29) = 7.07, p<0.0001). The liking ratings for objects on session one were related to responses and RTs during the memory recognition test on session 2. In a repeated measures logistic regression mixed effects model including only data for old objects seen on session 1, the odds of responding old multiplied for every unit increase in liking rating by 1.12 (95% CI [1.03 1.22], p=0.011). In a repeated measures linear regression mixed effects model, liking rating was weakly negatively related to RT ($\beta = -0.006$, 95% CI [−0.011–0.000], p=0.045).

