## [Decision Letter]

Thank you for submitting your article "A role for the hippocampus in value-based decisions: evidence from fMRI and amnesic patients" for consideration by *eLife*. Your article has been reviewed by three peer reviewers, including Thorsten Kahnt as the Reviewing Editor, and the evaluation has been overseen by Michael Frank as the Senior Editor. The following individuals involved in review of your submission have also agreed to reveal their identity: Lesley K Fellows (Reviewer #2); Sebastian Gluth (Reviewer #3).

The reviewers have discussed the reviews with one another and the Reviewing Editor has drafted this decision to help you prepare a revised submission.

Summary:

This manuscript describes two complementary experiments on the role of hippocampus in value-based decision making. The focus is on information sampling towards a bound which is hypothesized to rely on the hippocampus. The first study shows that fMRI responses in healthy human subjects correlate with response times in value-based but not perceptual tasks. This is interpreted as evidence for a role of hippocampus in retrieving information to guide value-based choice. The second study shows that patients with hippocampal lesions take longer for value-based choices and make more errors. This is interpreted as causal evidence for a role of hippocampus in value-base choice.

The reviewers agreed that the topic is of general interest and that the paper is clearly-written. They also agreed that it is a clear strength that both studies involve a perceptual choice task as a control, and that the key results are specific to value-based choice. Reviewers also identified a few major issues that should be addressed in a revised version. These include questions regarding alternative models or implementations of the drift diffusion model, the method used for multiple comparison correction of the fMRI data, and the unclear location and extent of the lesions in patients.

Essential revisions:

1) The authors fit a drift diffusion model (DDM) to the behavioral data. However, because no other model or parameterization is considered, it is unclear whether this model provides a particularly good account for the data. Arguably, a number of different models might provide a similar or perhaps even better description of the data. (For instance, the value data look like they would also be fit by a somewhat noisy process that is more-or-less flat except the extreme differences, i.e. the lowest and highest value difference choices are relatively fast. The extremes might be heuristics/rule-based (i.e. if X (my favorite) is available, always choose it; it Y is available, never choose it). It would be good if the authors would compare their model to at least one different class of models. In addition, the power-law transformation of the value difference in the drift rate and the collapsing bound parameters add flexibility to the DDM, but these choices were not justified on theoretical grounds or by means of a model comparison. The authors should consider a formal model comparison including simpler versions of the DDM (e.g. with plaw set to 1 and without collapsing bounds). In addition, a parameter recovery analysis of the winning model would be important to support the meaningfulness of the parameters. Alternatively, because it appears that the model is not essential to support the current conclusions, the authors may instead choose to significantly de-emphasize the focus on DDM throughout the paper.

2) Significance testing of the fMRI results was based on whole-brain cluster-wise correction. Examination of the whole-brain maps shows that the hippocampal activity modulated by RT is part of a very large cluster (53545 voxels), which ranges from occipital lobe through the whole brain up to orbitofrontal cortex, thereby including parahippocampal gyrus and hippocampus. Importantly, local peaks appear in separate brain regions, suggesting that the task activated several brain regions that are spatially distinct, but are treated as if they would come from the same cluster. Thus, the fact that the effect in the hippocampus is significant could be driven by signals in other parts of the brain. Given the anatomical hypothesis tested here, the reviewers think it would be more appropriate to perform correction within an anatomical ROI of the hippocampus (voxel-wise or cluster-wise). This way, only voxels within hippocampus can contribute to the significance of the effect. The reviewers also discourage the use of a mask of the striatum in Figure 4 to "mask out" other activations.

3) It would be important to include anatomical brain images from the patient population. These should show the extent and overlap of the lesions, and show whether the damaged regions overlap with the hippocampal region identified in the fMRI study.

4) Finally, it is unclear why the PPI analysis used a median split for RT rather than a parametric modulation. The gPPI package used here allows parametric regressors to be used as the psychological variable to construct PPI regressors. This approach would be preferable, as it would allow a more direct test of the hypothesis that hippocampus-striatum connectivity is correlated with RT.

[Editors' note: further revisions were requested prior to acceptance, as described below.]

Thank you for resubmitting your work entitled "The hippocampus supports deliberation during value based decisions" for further consideration at *eLife*. Your revised article has been favorably evaluated by Michael Frank as the Senior Editor, and three reviewers, one of whom is a member of our Board of Reviewing Editors.

The manuscript has been improved but there are some remaining issues that need to be addressed before acceptance, as outlined below:

As you can see, reviewer #3 makes additional suggestions, which all reviewers agree should be considered. In particular, this reviewer points out that the parameter recovery analysis (an Essential point in the initial review) was not included in the revised manuscript and also encourages you to include the model comparison in the manuscript (point 1). Moreover, this reviewer makes suggestions for how to approach the model comparison (point 2), which you may want to consider. Finally, and most importantly, the Abstract should be changed to better reflect the findings (point 3).

*Reviewer #1:*

The authors have adequately addressed all essential points.

*Reviewer #2:*

The authors have been very thorough and thoughtful in their response to reviews. This version of the paper is well-argued and very interesting. I have no further comments or modifications to suggest. I recommend it for publication. Thanks for giving me the opportunity to serve as a reviewer.

*Reviewer #3:*

Bakkour and colleagues have done a quite good job addressing most of the reviewers' points, in particular the concerns about the multiple-comparison correction, the lesion information of patients, the analysis of chosen value signals, and the PPI analysis.

However, there are still some issues with the revised manuscript, so that we do not think that it can already be accepted in its current state.

1) In their rebuttal, the authors state that they compared their DDM version to simpler versions without power transformation and without a collapsing bound and found that (at least on the group level) their DDM version wins this comparison. However, none of this is reported in the manuscript, and we are wondering why. We strongly suggest that these comparisons should be reported, in particular, because the more conceptual justifications that are given are not convincing (e.g., the authors say "there is no reason to assume that the relationship [between value and drift] is linear"; speaking with Occam's Razor, we'd say "there is no reason NOT to assume a linear relationship"). Furthermore, the authors do not report a parameter recovery analysis that was asked for in the previous essential revisions.

2) The authors added a comparison of the DDM with a heuristic model. In general, we appreciate this interesting comparison. However, there are two issues that we have with this comparison.

First, for the heuristic model the authors draw RTs from a normal distribution. The authors may consider verifying whether the RTs in their task were indeed normally distributed. This may be the case given the 3 sec time limit, but often RTs are skewed. In that case, the authors may consider more appropriate distributions such as log-normal or ex-Gaussian to make the model comparison fair.

Second, the authors appear to struggle with the quantitative comparison between the two models, because they do not have a good error model for what they call "trivial" decisions. There is a simple (and very common) solution to this problem. Instead of punishing errors in "trivial" decisions by assuming a probability of p =.01, one would treat this probability p as a free parameter (i.e., assuming a so-called "trembling hand" error rate of p in trivial decisions). This "trembling hand" error is very common in research on strategy selection (e.g., Rieskamp and Otto, 2006, JEP General). Allowed values for p might be restricted to some plausible range (e.g., p <.1). This would render calling the comparison an "unsatisfactory exercise" unnecessary. Note, however, that we still think that the (also very interesting) qualitative comparison should remain in the paper, too.

3) Looking at the new ROI analyses reported in Figure 3—figure supplement 3E-H (and the associated Table 3—source data 8), we saw that there is actually hippocampal activity related to RT in the perceptual task. We think that this requires rephrasing the Abstract, in which it is stated that this relationship would be "not observed in a perceptual decision task". Instead, the significantly stronger relationship in value-based compared to perceptual decision making should be emphasized.

---

## [Author Response]

Essential revisions:1) The authors fit a drift diffusion model (DDM) to the behavioral data. However, because no other model or parameterization is considered, it is unclear whether this model provides a particularly good account for the data. Arguably, a number of different models might provide a similar or perhaps even better description of the data. (For instance, the value data look like they would also be fit by a somewhat noisy process that is more-or-less flat except the extreme differences, i.e. the lowest and highest value difference choices are relatively fast. The extremes might be heuristics/rule-based (i.e. if X (my favorite) is available, always choose it; it Y is available, never choose it). It would be good if the authors would compare their model to at least one different class of models. In addition, the power-law transformation of the value difference in the drift rate and the collapsing bound parameters add flexibility to the DDM, but these choices were not justified on theoretical grounds or by means of a model comparison. The authors should consider a formal model comparison including simpler versions of the DDM (e.g. with plaw set to 1 and without collapsing bounds). In addition, a parameter recovery analysis of the winning model would be important to support the meaningfulness of the parameters. Alternatively, because it appears that the model is not essential to support the current conclusions, the authors may instead choose to significantly de-emphasize the focus on DDM throughout the paper.

We thank the reviewers for this thoughtful comment. The last statement is most consistent with our perspective. As the reviewers note, the particular model we used to fit the behavioral data is not essential to the conclusions of this study. We had tried to de-emphasize the diffusion model in the original submission and we have buttressed this position substantially in our revision. However, as we do show fits to choice-RT data, we felt we should also address the points concerning parameterization and alternative models. The latter turned out especially constructive. Most of these points are reflected in changes in the manuscript.

a) Model justification. Variants of the bounded drift diffusion models have been applied successfully to both perceptual- and value-based choice-response time experiments. Our study connects to the prior literature on value-based decisions (Milosavljevic et al., 2010; Krajbich et al., 2010; Sokol-Hessner et al., 2012). Thus, it seemed essential that readers recognize that the data are comparable to previous studies. Further, for the perceptual task, we would be remiss if we were to neglect diffusion models. In short, the previous literature and the desire to compare to perception motivated the use of sequential sampling with option stopping (SSwOS) models. We now state clearly that the use of these models does not commit one to the interpretation that the choice and RT are reconciled by an accumulation of noisy evidence to a threshold (Discussion).

b) Parameterization. The reviewers raised legitimate questions about our use of a power law transformation of *∆Value* and the use of collapsing bounds. As a rule, we wished to maintain similar strategies in fitting perception and preference data, but not at the expense of inducing a poor fit to the latter. The power law transformation allows us to find a suitable monotonic mapping between *∆Value* and drift rate without overfitting (e.g., a different drift rate for each *∆Value* or *∆Value*-quantile). Importantly, there is no reason to assume that the relationship is linear. On theoretical and empirical grounds, the mapping should be approximately linear for random dot motion (Shadlen et al., 2006), and the same considerations ought to apply to our version of the stochastic color display too. However, the assumption of linearity is overly restrictive for ∆V and could penalize spuriously the fits to these data. Model comparison at the combined level demonstrates this to be so. We could not reject an exponent of unity for individual subjects, but the analysis is underpowered (~210 trials per subject). We now explain this in the Materials and methods.

The collapsing bounds are necessary to account for RT distributions and to explain the mean RT on error trials. This is the normative implementation of bounded drift diffusion when there are multiple difficulty levels (Drugowitsch et al., 2012). The shape of the collapse depends on the set of difficulty levels and time costs. We allowed the same degree of freedom to the snacks data. Model comparison favored a collapsing bound for the aggregate data (BIC=149).

c) An alternative [heuristic] model. We implemented a model like the one suggested by the reviewer, which posits the following. A subset of food items are valued as either highly desirable (D+) or highly undesirable (D−). The remaining items are in the moderate (D≈) category. This yields three types of decisions, which should behave as follows based on the alternative heuristic model: (i) Decisions between D+ and an item from D− or D≈ should produce a fast choice of the D+ item regardless of ∆Value. (ii) Decisions between D− and an item from D≈ should produce a fast choice of the D≈ item regardless of ∆Value. (iii) Decisions between two items from the same class (D+ or D− or D≈) may be stochastic (described by a logistic function of ∆Value) and slow, regardless of ∆Value. We refer to i and ii as *trivial* decisions and to iii as *non-trivial* decisions.

This model can be fit straightforwardly to the data, but there is one thorny issue. According to the logic of the heuristic model, *trivial* decisions should not produce errors, whereas in the data these errors were common (see Figure 5—figure supplement 2 insets). This observation already raises skepticism about the model, but it does not rule it out. For these error trials, we adjust the fitting procedure to penalize them severely (see revised Materials and methods). The comparison favors diffusion for the participants in Experiment 1 and for the healthy controls in Experiment 2. It is less conclusive for the amnesic patients. However, the assignment of penalty on the *trivial* errors renders model comparison unsatisfying.

We therefore adopted a qualitative comparison. Under the heuristic model, for the bulk of trials in the *non-trivial* category, RT should not depend on |∆Value|. This is demonstrably false for participants in Experiment 1 and for the healthy controls in Experiment 2. However, it is true for the amnesic patients, and the difference between the patients and the controls is statistically reliable. We have added Figure 5—figure supplement 2, which shows this contrast between patients and healthy controls. The addition of this analysis bolsters a point that was evident by eye in the original submission and therefore improves the paper by further characterizing the difference between patients and controls. It also serves the general goal of further deemphasizing the literal interpretation of diffusion.

2) Significance testing of the fMRI results was based on whole-brain cluster-wise correction. Examination of the whole-brain maps shows that the hippocampal activity modulated by RT is part of a very large cluster (53545 voxels), which ranges from occipital lobe through the whole brain up to orbitofrontal cortex, thereby including parahippocampal gyrus and hippocampus. Importantly, local peaks appear in separate brain regions, suggesting that the task activated several brain regions that are spatially distinct, but are treated as if they would come from the same cluster. Thus, the fact that the effect in the hippocampus is significant could be driven by signals in other parts of the brain. Given the anatomical hypothesis tested here, the reviewers think it would be more appropriate to perform correction within an anatomical ROI of the hippocampus (voxel-wise or cluster-wise). This way, only voxels within hippocampus can contribute to the significance of the effect. The reviewers also discourage the use of a mask of the striatum in Figure 4 to "mask out" other activations.

We completely agree and are grateful for the suggestion. Indeed, our hypotheses were centered on the hippocampus to begin with and therefore we now provide the results of the fMRI analyses in an anatomical mask of bilateral hippocampus (from the Harvard-Oxford atlas). This ROI-focused analysis confirms the results previously reported from the whole-brain analysis, showing that BOLD activity within the hippocampal anatomical ROI correlates more positively with RT during the value-based decision task than the perceptual task. The conjunction between this contrast and the memory contrast of Hits vs. Correct rejections also replicates when restricting the analysis to the hippocampal ROI. We have updated Figure 3—figure supplement 3 and the activation table in Figure 3—source data 8 in the updated manuscript accordingly.

We appreciate the reviewers’ comment regarding “masking out” activations in main figures. In addition to the new PPI analysis using RT as a continuous measure (see response to comment 4 below), we present unmasked whole-brain-corrected activations in updated Figure 4 in the revised manuscript.

3) It would be important to include anatomical brain images from the patient population. These should show the extent and overlap of the lesions, and show whether the damaged regions overlap with the hippocampal region identified in the fMRI study.

Thank you, we agree that the anatomical specificity is important to display where possible. These patients have been reported on in several studies before and we now include the available scans and quantification of their lesions here. Specifically, for five of the six patients, clinical and/or research scans were available, which indicated medial temporal lobe lesions including the hippocampus. As described in the subsection “Experiment 2: amnesic patients and age-matched healthy control participants”, two patients had lesions restricted to the hippocampus, while 3 patients had volume loss additionally extending outside of the hippocampus. A new figure is now included depicting the lesion in each of these patients (Figure 5—figure supplement 3). Two patients for whom only clinical scans are available show extensive lesion (P02) and hyperintensities (P05), respectively, along the entire long axis of the hippocampus, including the mid-hippocampal level that shows peak activation on the whole-brain-corrected level in the fMRI study. Two patients for whom research MRIs are available (P03, P06) similarly showed that their lesion includes the locus of peak fMRI activation. For P01 this determination could not be made given that only a clinical CT scan was available. We provide this individual lesion data in the revised manuscript together with the individual behavioral data (as before), allowing the reader to visually assess how differences in lesion extent may relate to differences in behavior. For example, the reader might see that P06, compared to the others, seems to have relatively less damage to the hippocampus and also has the steepest choice and RT curves among patients. Conversely, P02 seems to have the most damage to the hippocampus and has among the shallowest choice and RT curves among patients.

4) Finally, it is unclear why the PPI analysis used a median split for RT rather than a parametric modulation. The gPPI package used here allows parametric regressors to be used as the psychological variable to construct PPI regressors. This approach would be preferable, as it would allow a more direct test of the hypothesis that hippocampus-striatum connectivity is correlated with RT.

We appreciate the reviewers’ suggestion to run the PPI with RT as a parametric regressor, rather than splitting RT along the median, as this is indeed more consistent with the other analyses and our view of the underlying relationship between choice and RT. We now report the results of this new analysis in the subsection “Connectivity between hippocampus and striatum increases with value-based decision time”. As shown, the parametric analysis reveals a significant result in the striatum, with the main cluster in the caudate. The whole-brain corrected map is presented in an updated Figure 4 in the revised manuscript. The full uncorrected map is available here: https://neurovault.org/collections/BOWMEEOR/images/129376. The Materials and methods section as well as the associated activation table (Figure 4—source data 1) were all updated to reflect this change.

[Editors' note: further revisions were requested prior to acceptance, as described below.]

Reviewer #3:

Bakkour and colleagues have done a quite good job addressing most of the reviewers' points, in particular the concerns about the multiple-comparison correction, the lesion information of patients, the analysis of chosen value signals, and the PPI analysis.However, there are still some issues with the revised manuscript, so that we do not think that it can already be accepted in its current state.1) In their rebuttal, the authors state that they compared their DDM version to simpler versions without power transformation and without a collapsing bound and found that (at least on the group level) their DDM version wins this comparison. However, none of this is reported in the manuscript, and we are wondering why. We strongly suggest that these comparisons should be reported, in particular, because the more conceptual justifications that are given are not convincing (e.g., the authors say "there is no reason to assume that the relationship [between value and drift] is linear"; speaking with Occam's Razor, we'd say "there is no reason NOT to assume a linear relationship"). Furthermore, the authors do not report a parameter recovery analysis that was asked for in the previous essential revisions.

We have incorporated these suggestions in the revised manuscript. We now report the comparison of the full model to ones with combinations of flat bounds and no power law (subsection “Behavior in both decision tasks conforms to sequential sampling models”). We also performed a parameter recovery analysis for the full model. This analysis is reported in Figure 2—figure supplement 3. The result is reassuring, and we explain why recovery is less than stellar in the two instances that this occurs. The parameterization of the bound contains *d.f.* that approximates a normative solution (Drugowitsch et al., 2012) but the parameters are of lower dimension if the bound is simpler.

In case helpful, we would like to share our thoughts in response to R3’s wonder at why we did not include the model comparison in the main text in our first revision. Our reasoning is best captured by the reviewers’ statement: “…because it appears that the model is not essential to support the current conclusions, the authors may instead choose to significantly de-emphasize the focus on DDM throughout the paper.” We found this comment helpful and felt it accurately reflects our position. We therefore revised accordingly, aiming to emphasize that the contributions of the paper do not rely on a strict interpretation of the model. We performed the fits to establish a connection to previous literature and to facilitate parallel exposition with the perceptual control. We performed a model comparison to fully respond to the reviewer’s request to see the comparisons, but we chose not to include the formal comparison in the revised manuscript because we felt it would run counter to our goal to deemphasize the model throughout the paper. A capacity to fit the data with a diffusion model does not necessarily implicate a mechanism of bounded noisy-evidence accumulation, and we were concerned that undue (in our view) emphasis on model parameters could convey a confusing and potentially misleading message, as it seems to imply that we take the fits more seriously than we do. All that said, we accept that there may be readers with expertise in this area, like R3, who find the omission puzzling, and who would benefit from seeing the model comparison. Therefore, we have now included this information in the subsection “Behavior in both decision tasks conforms to sequential sampling models” and in the legend of the new Figure 2—figure supplement 3.

Finally, for the reviewer, we include Author response image 1 that shows the relationship between drift rate and ∆Value quantile. It conforms to the same power law as the one identified by our fitting procedure. A linear function of stimulus strength is exceptional, which is why many practitioners fit a different drift rate to each difficulty level.

**Author response image 1. respfig1:** Justification for use of power law. The red points are the fitted values of *κ* (Eq. 2) computed for each of the five non-zero quantiles of |∆Value|. Error bars are s.e. The dashed line is the best fitting line on the log-log graph, which yields a slope of 0.68 ± 0.06. The quantiles are the same as those used in Figure 2.

2) The authors added a comparison of the DDM with a heuristic model. In general, we appreciate this interesting comparison. However, there are two issues that we have with this comparison.First, for the heuristic model the authors draw RTs from a normal distribution. The authors may consider verifying whether the RTs in their task were indeed normally distributed. This may be the case given the 3 sec time limit, but often RTs are skewed. In that case, the authors may consider more appropriate distributions such as log-normal or ex-Gaussian to make the model comparison fair.

The heuristic model assumes that the RT is either fast or slow, so it assumes that the distribution of RTs across all trials will be bimodal. Using two Gaussian distributions with different means and standard deviations is consistent with this assumption. For purposes of model comparison, it would not be appropriate to adopt the shape of the distribution from the data. Diffusion models supply a generative model for the RT distributions. We have supplied a generative model for RT distributions consistent with the trivial/nontrivial heuristic. While one might consider other generative models, for the sake of parsimony we think it would be best to stay clear of this rabbit hole.

Second, the authors appear to struggle with the quantitative comparison between the two models, because they do not have a good error model for what they call "trivial" decisions. There is a simple (and very common) solution to this problem. Instead of punishing errors in "trivial" decisions by assuming a probability of p =.01, one would treat this probability p as a free parameter (i.e., assuming a so-called "trembling hand" error rate of p in trivial decisions). This "trembling hand" error is very common in research on strategy selection (e.g., Rieskamp and Otto, 2006, JEP General). Allowed values for p might be restricted to some plausible range (e.g., p <.1). This would render calling the comparison an "unsatisfactory exercise" unnecessary. Note, however, that we still think that the (also very interesting) qualitative comparison should remain in the paper, too.

We have implemented the reviewer’s suggestion. This parameterization of the heuristic model still provides a poorer account for the data compared to DDM (BIC = 425.88). We discuss this in the third paragraph of the subsection “Heuristic model”.

3) Looking at the new ROI analyses reported in Figure 3—figure supplement 3E-H (and the associated Table 3—source data 8), we saw that there is actually hippocampal activity related to RT in the perceptual task. We think that this requires rephrasing the Abstract, in which it is stated that this relationship would be "not observed in a perceptual decision task". Instead, the significantly stronger relationship in value-based compared to perceptual decision making should be emphasized.

Thank you. We have corrected this oversight.